# Genetic Variation, DIMBOA Accumulation, and Candidate Gene Identification in Maize Multiple Insect-Resistance

**DOI:** 10.3390/ijms24032138

**Published:** 2023-01-21

**Authors:** Yining Niu, Xiaoqiang Zhao, Wun Chao, Peina Lu, Xiaodong Bai, Taotao Mao

**Affiliations:** 1State Key Laboratory of Aridland Crop Science, College of Agronomy, Gansu Agricultural University, Lanzhou 730070, China; 2United States Department of Agriculture, Fargo, ND 58102-2765, USA

**Keywords:** maize, DIMBOA accumulation, Asian corn borer/corn leaf aphid, genetic diversity, association mapping, general linear model, mixed linear model, pleiotropy, candidate genes

## Abstract

Maize seedlings contain high amounts of 2,4-dihydroxy-7-methoxy-1,4-benzoxazin-3-one (DIMBOA), and the effect of DIMBOA is directly associated with multiple insect-resistance against insect pests such as Asian corn borer and corn leaf aphids. Although numerous genetic loci for multiple insect-resistant traits have been identified, little is known about genetic controls regarding DIMBOA content. In this study, the best linear unbiased prediction (BLUP) values of DIMBOA content in two ecological environments across 310 maize inbred lines were calculated; and their phenotypic data and BLUP values were used for marker-trait association analysis. We identified nine SSRs that were significantly associated with DIMBOA content, which explained 4.30–20.04% of the phenotypic variation. Combined with 47 original genetic loci from previous studies, we detected 19 hot loci and approximately 11 hot loci (in Bin 1.04, Bin 2.00–2.01, Bin 2.03–2.04, Bin 4.00–4.03, Bin 5.03, Bin 5.05–5.07, Bin 8.01–8.03, Bin 8.04–8.05, Bin 8.06, Bin 9.01, and Bin 10.04 regions) supported pleiotropy for their association with two or more insect-resistant traits. Within the 19 hot loci, we identified 49 candidate genes, including 12 controlling DIMBOA biosynthesis, 6 involved in sugar metabolism/homeostasis, 2 regulating peroxidases activity, 21 associated with growth and development [(auxin-upregulated RNAs (SAUR) family member and v-myb avian myeloblastosis viral oncogene homolog (MYB)], and 7 involved in several key enzyme activities (lipoxygenase, cysteine protease, restriction endonuclease, and ubiquitin-conjugating enzyme). The synergy and antagonism interactions among these genes formed the complex defense mechanisms induced by multiple insect pests. Moreover, sufficient genetic variation was reported for DIMBOA performance and SSR markers in the 310 tested maize inbred lines, and 3 highly (DIMBOA content was 402.74–528.88 μg g^−1^ FW) and 15 moderate (DIMBOA content was 312.92–426.56 μg g^−1^ FW) insect-resistant genotypes were major enriched in the Reid group. These insect-resistant inbred lines can be used as parents in maize breeding programs to develop new varieties.

## 1. Introduction

Maize (*Zea mays*), is an important agro-economical crop that is utilized globally as food, animal feed, and biofuel products with production of more than 1.14 billion tons in 2018 [1]. Despite the large production area for maize, extensive losses are common in China due to a range of biotic stressors, such as multiple insect injury [1]. Over 350 insect pests have been identified for maize. Of these, Asian corn borer (ACB) (*Ostrinia furnacalis*; Lepidoptera, Pyralidae) is one of the most destructive insect pests in maize. Newly hatched ACB larvae primarily feed on leaves during the whorl stage; subsequently, the 3rd or 4th instars bore into the stalk. This may cause yield losses of 10 to 30% in the outbreaks recorded in China [2]. In addition, the aboveground parts of maize are susceptible to corn leaf aphid (CLA) (*Rhopalosiphum maidis*; Homoptera, Aphididae), especially in tropical and warmer temperature regions [3,4]. CLA infestation in maize seedlings slows down plant development, reduces plant height, and decreases grain yield [5]. CLA damage can even occur through the maize tassel in which the accumulation of sticky honeydew can prevent the shedding of pollen, with yield losses of up to 90% [4,6]. Several aphid species can also transmit maize dwarf mosaic virus in a non-persistent manner [7], causing yield losses as high as 70% [8].

Chemical insecticides are widely used to reduce yield losses in maize incurred by insect pests, including ACB and CLA. However, residues from these insecticides can pose a significant health hazard to maize consumers and also cause harm to the environment [9,10]. Furthermore, the overuse of pesticides may also result in the development of chemical resistance to insects and the emergence of secondary pests [9,11]. Fortunately, maize is a genetically diverse crop that exhibits a wide variation in its resistance to multiple insects [9,12,13]; the varieties of maize that show high resistance to insect pests can be used to thwart multiple insect attacks via their various defense mechanisms [9,14]. In maize, the chemical defense metabolite 2,4-dihydroxy-7-methoxy-1,4-benzoxazin-3-one (DIMBOA), a cyclic hydroxamic acid derivative (benzoxazinoids, bxs), is known to confer resistance to leaf-feeding by both ACB [2,15] and CLA [4,16] via inhibitory and toxicity mechanisms. Previous studies reported that the DIMBOA could constitute > 1% of the dry weight of maize seedlings [17] and could be more abundant in adult maize plants after the induction of defense mechanisms [18]. The advances in molecular biology and statistical methods over recent years have led to the identification of multiple quantitative trait loci (QTLs), associated markers, and candidate genes for maize resistance to leaf-feeding by different corn borers and aphids [2,4,12,19,20]. However, only one study reported the involvement of QTLs responsible for DIMBOA content [15], and the low resolution of QTL linkage mapping methodology did not provide a convincing conclusion that genetic variation in maize resistance to multiple insect pests was associated with DIMBOA content and bxs alleles.

Association mapping based on linkage disequilibrium (LD) provides a fine-mapping method that enables researchers to identify functional variation in a broader germplasm background [21]. Both QTL linkage mapping and association mapping methods could be complementary strategies for investigating genetic variation in maize related to DIMBOA content. At present, only a few maize genotypes (Mc37, Mp708, Mp704, D06, CML103, A637, and EP39) are known to be ACB-resistant germplasms [20]; little is known with regard to multiple insect-resistance breeding in maize. Therefore, there is an urgent need to screen genetic resources and characterize the mechanisms associated with multiple insect resistance. The testing of inbred maize lines in terms of their DIMBOA levels in response to ACB and CLA feeding may serve as a potent approach to evaluate and develop new varieties with multiple insect-resistance. Gansu Province in China is one of the core bases for maize seed production and breeding, and a large number of elite inbred lines have been cultivated over the last decade. However, the development of breeding programs for multiple insect-resistance has been slow in this region. For this reason, we collected and investigated 310 elite inbred maize lines with broad genetic backgrounds from different ecological environments in Gansu Province, China. Our objectives were to (i) evaluate the accumulation of DIMBOA in the V6 stage of these different genotypes in different ecological environments and to identify elite multiple insect-resistant parents for use in breeding programs; (ii) explore the genetic diversity and population structure of these maize lines by 186 polymorphic simple repeated sequences (SSRs) in the 10 maize chromosomes, and (iii) identify SSRs that were significantly associated with DIMBOA content and the best linear unbiased prediction (BLUP) values of DIMBOA content in two ecological environments via a general linear model (GLM) and a mixed linear model (MLM). We also detected hot genetic loci and candidate genes associated with multiple insect-resistance by combining our research findings with those of previous studies. These findings have laid a foundation for further multiple insect-resistance marker-assisted selection (MAS) breeding in maize.

## 2. Results

### 2.1. DIMBOA Content Variation

The maize-feeding insects begin to damage maize leaves at the V5-V6 seedlings stage in the field [15]. At the V6 stage, sufficient variation was observed for DIMBOA content in 310 maize inbred lines in both Zhangye (E1) and Longxi (E2) ecological environments (Figure 1A). The average DIMBOA content of 310 inbred lines seedlings in the E1 environment ranged from 8.66 (T58) to 528.88 μg g^−1^ FW (ZY19-Jiu1101) with an overall mean of 94.21μg g^−1^ FW; similarly, the average DIMBOA content of these seedlings in the E2 environment varied from 8.84 (T58) to 493.40 μg g^−1^ FW (RX20-1006) with an overall mean of 93.26 μg g^−1^ FW (Figure 1B). Moreover, factorial analysis of variance also proved significant for the tested variables (genotypes and environments) and their interaction with the DIMBOA content in the seedlings of these maize genotypes (Figure 1C). These data indicated that the DIMBOA formation and accumulation were controlled by a number of factors, including maize’s genetic constitutions, external environments, and their interaction. Therefore, after maize-feeding insects, such as ACB and CLA feed on leaves, DIMBOA can be induced rapidly in maize to confer resistance to these insects, along with other adverse responses, such as anorexia and toxic reactions in these insects.

In addition, the DIMBOA content of these maize materials in the two ecological environments followed a typical partial distribution, as the skewness and kurtosis values of the DIMBOA content were > 1.0 (Figure 1B). We speculated that this phenomenon may be related to the unequal enrichment of DIMBOA content in the selected maize inbred lines. Further analysis showed that the estimated broad-sense heritability ($H_{B}^{2}$) and genotype × environment interaction heritability ($H_{GE}^{2}$) of DIMBOA content in all inbred lines across both ecological environments were 74.64 and 37.32%, respectively (Figure 1D). The data demonstrated that DIMBOA content was less affected by the environment when compared to other insect-resistant traits, especially the leaf feeding rating of corn borer (LFR) and the number of holes of corn borer (HO), and their $H_{B}^{2}$ were 37.5 and 46.4% [2], respectively. Meanwhile, the genetic variation coefficient (CVg) of DIMBOA content among 310 maize inbred lines in both E1 and E2 environments were 108.47 and 106.22%, respectively (Figure 1B). Thereby, it is necessary to detect the genetic loci responsible for DIMBOA content in maize. 

### 2.2. Genetic Diversity and Population Structure Analysis

Genetic diversity and population structure analysis are important tools for germplasm characterization and subsequent utilization in multiple insect-resistance improvements. A total of 186 polymorphic SSRs were used to analyze the distribution of 748 alleles in 310 maize inbred lines. The number of alleles varied from two to eight at a locus. The umc1917 (Bin 1.04, ctg14), umc2314 (Bin 6.01, ctg268) and umc2031 (Bin 8.06, ctg361) exhibited the maximum number of alleles (8 alleles) (Appendix A). It is well known that the marker attributes, i.e., the polymorphism information content (PIC) value and Shannon-Wiener’s index (I) value are routinely used to evaluate the informativeness of the primers. In this study, the PIC value of each SSR ranged between 0.294 (umc2297, Bin 5.03, ctg220) and 0.826 (umc2314, Bin 6.01, ctg268). Out of 186 SSRs, 69 markers were highly polymorphic, with a PIC ≥ 0.600 (Appendix A). Similarly, the average I value was 1.359 and ranged between 0.585 (umc2112, Bin 1.04, ctg21; umc2043, Bin 10.05, ctg415) and 2.410 (umc2314, Bin 6.01, ctg268) (Appendix A). These data showed that a PIC value ≥ 0.600 was observed in approximately 40% of the markers, suggesting that these SSRs were very informative and useful in the assessment of genetic diversity, population structure, and marker-DIMBOA content association analysis.

In addition, the population structure of these inbred lines was analyzed by STRUCTURE software. When various groups (K value) ranging from 1 to 12 were compared, the ΔK reached the maximum value when K = 5, thus the 310 maize inbred lines were divided into five optimal groups (Figure 2A). The inbred lines with a membership probability of ≥ 0.500 were assigned to the same groups, and if the inbred lines had a membership probability of less than this value, they were assigned to a mixed group (not assigned to any of the five groups) [22,23]. Of the 310 inbred lines, 294 (accounting for 94.84%) were assigned to either one of the five groups, including the Lüda red cob (LRC; 44 lines) group, Tang si ping tou (TSPT; 81 lines) group, Lancaster (Lan; 52 lines) group, P (47 lines) group, and Reid (70 lines) group, respectively, and the remaining 16 inbred lines (accounting for 5.16%) were categorized as a mixed group (Figure 2B and Figure 3A; Appendix A).

### 2.3. Evaluation of Multiple Insect-Resistance of Inbred Lines and Dissection of Their Attributive Groups

According to the performance of DIMBOA content among all inbred lines in the two ecological environments, these germplasms were divided into five types (type I to type V). Type I (DIMBOA content in E1/E2: 467.11–528.88/402.74–493.40 μg g^−1^ FW; accounted for 0.97%) had three genotypes, which were 19LX-1230, RX20-1006, and ZY19-Jiu1101; they were defined as high insect-resistant germplasms. Type II (DIMBOA content in E1/E2: 312.92–426.56/321.41–424.15 μg g^−1^ FW; accounted for 4.84%) had 15 inbred lines including F1227, M1005, F0501, LongF1008, 1201, M1009, PH1CRW, 1512, F3202, F1220, F3208, F2260, F1233, F2303, and F3210, and they were defined as moderate insect-resistant germplasms. Type III (DIMBOA content in E1/E2: 184.88–292.42/176.52–296.96 μg g^−1^ FW; accounted for 8.71%) had 27 genotypes, including ShanM3304, M1001, MeizaS2–3, Jizaoyu, M0803, F0306, M10202, ly6305, F2211, M1202, F0311, M1409Ying, F1501, M0124, M0822, M0105, F2502, M0505, PHHJC, 8723-2, M0306, F2222, F2213, M0824, M0125, 747, and F3226, and they were defined as insect-resistant germplasms. Type IV (DIMBOA content in E1/E2: 103.88–184.62/99.76–187.68 μg g^−1^ FW; accounted for 16.77%) included 52 moderate insect-susceptible germplasms. Type V (DIMBOA content in E1/E2: 8.66–101.14/8.84–99.40 μg g^−1^ FW; accounted for 68.71%) included 213 insect-susceptible germplasms (Figure 3C). Furthermore, the three high insect-resistant inbred lines and 15 moderate insect-resistant inbred lines belonged to the Reid (accounted for 50.00%), LRC (accounted for 22.22%), P (accounted for 16.67%), and Lan (accounted for 11.11%) groups, respectively (Figure 3B). The data suggested that Reid was an important insect-resistant group, and more attention should be given to the improvement of new insect-resistant varieties using Reid genotypes.

### 2.4. Association Analysis of DIMBOA Content and BLUP Values

In this study, we retrieved the genetic distance (centimorgan, cM) of 186 SSRs on an IBM2 2008 Neighbors map frame (https://www.maizegdb.org/data_center/map (accessed on 18 September 2022)) (Appendix A) to build a genetic linkage map, which spanned a total length of 6684.4 cM (Figure 4). Then, the associated SSR loci of DIMBOA content in both ecological environments and the BLUP values were assessed using Tassel 3.0 software with GLM (Q) and MLM (Q + K) approach. We detected nine and three significant (*p* < 0.01) SSR loci associated with DIMBOA content in both environments (E1 and E2) by GLM and MLM, respectively; these SSR loci were in Bin 1.04, Bin 1.11, Bin 2.01, Bin 4.00, Bin 4.01, Bin 6.02, Bin 8.04, and Bin 10.04. The phenotypic variation explained by these SSR loci ranged from 4.30% (umc2363, Bin2.01, ctg69) in the E1 environment to 20.04% (umc1008, Bin 4.00, ctg154) in the E2 environment via GLM, and ranged from 4.74% (umc1858, Bin 8.04, ctg349) in the E2 environment to 10.57% (umc1017, Bin 4.01, ctg155) in the E1 environment via MLM (Figure 4; Table 1). Likewise, a total of six and two SSR loci in Bin 1.04, Bin 1.11, Bin 4.00, Bin 4.01, and Bin 10.04 were identified to be significantly (*p* < 0.01) associated with BLUP values by GLM and MLM, respectively, which explained 5.41% (umc1054, Bin 10.04, ctg412)–19.42% (umc1017, Bin 4.01, ctg155) by GLM, and explained 9.29% (umc1917, Bin 1.04, ctg14)–13.70% (umc1008, Bin 4.00, ctg154) by MLM (Figure 4; Table 1).

### 2.5. Identification of Hot Genetic Loci and Candidate Genes Associated with Multiple Insect-Resistant Traits

Next, we attempted to obtain hot genetic loci for multiple insect-resistant traits of ACB and CLA to lay a foundation for fine mapping and candidate gene prediction, verification, and breeding application. First, we collected 47 original genetic loci for ACB-/CLA-resistant traits including DIMBOA content, tunnel length of corn borer (TL), aphid incidence rate (AIR), average aphid incidence grade (AIG), LFR, HO, tunnel length/number of holes of corn borer (TL/HO), and aphid resistance (AR) from previous studies (Table 2), and combined our results (Table 1) to construct a physical map (Figure 5). Further, 19 hot genetic loci (Loci 1-Loci 19) for multiple insect resistance were identified on 8 chromosomes except chromosomes 3 and 7, resulting in the identification of 49 candidate genes, including 12 controlling DIMBOA biosynthesis, 6 involved in sugar metabolism/homeostasis, 2 participating in peroxidases (POD) activity, 9/12 associated with auxin-upregulated RNAs (SAUR) family member/ v-myb avian myeloblastosis viral oncogene homolog (MYB), and 3/2/1/1 responsible for the lipoxygenase/cysteine protease/restriction endonuclease/ubiquitin conjugating-enzyme (Figure 5; Table 3; Appendix A); these candidate genes may be utilized to determine the multiple insect-resistance to ACB and CLA through their complex interaction network.

### 2.6. Expression Levels of Five Candidate Genes Responsible for DIMBOA Biosynthesis

We randomly selected 5 of the 12 candidate genes that control DIMBOA biosynthesis, including *GRMZM2G085381* (*bx1*), *GRMZM6G617209* (*bx6*), *GRMZM2G441753* (*bx7*), *GRMZM2G311036* (*bx10*), and *GRMZM2G336824* (*bx11*) to examine their relative expression levels in maize seedlings of RX20-1006 (high insect-resistant line) and T58 (insect-susceptible line) in the E1 environment at the V6 stage using real-time PCR (RT-qPCR). The results showed that the relative expression levels of the five genes were significantly correlated with the DIMBOA content in the two maize genotypes (Figure 6A–C).

## 3. Discussion

### 3.1. Defense Strategies of Maize against ACB-/CLA-Feeding

It is well known that the plethora of ACB and CLA that either simultaneously or concurrently attack multiple maize parts, such as newly hatched ACB feeds on whorl leaves and later instars, tunnel into the stalk or the ear to feed on pith tissues or fresh kernels [25], resulting in a reduction of photosynthetic property, disruption of nutrient and water transport, an increase of stalk lodging and bacterial/fungal infections [25,26], and ultimately complicates harvesting practices and reduces grain yield and quality [27,28]. In addition, CLA is a phloem sap-sucking pest [29], and it can absorb nutrients from phloem sap and alter source-sink patterns [30]; its digestive waste products, i.e., honeydew, can also deposit on the leaf surface of maize and promote mold growth [6]. CLA is also a vector for some plant viral diseases that facilitate pathogen entry and cause maize leaves to curl, discolor, and wilt after serious CLA infestations [6].

In response to insect attack, maize has evolved an extensive array of defense strategies to prevent ACB or CLA feeding and colonization. Increasing evidence has verified that maize-derived compounds bxs, e.g., 2,4-dihydroxy-7-1,4-benzoxazin-3-one glucoside (DIMBOA-Glc), DIMBOA, and 6-methoxy-3h-1,3-benzoxazol-ne (MBOA) are multifunctional defense metabolites that can protect maize against insect pests feeding and pathogens [31,32,33]. DIMBOA acts as a feeding deterrent in maize that can decrease in vivo endoproteinase activity in the larval midgut of the European corn borer, thus limiting the availability of amino acids, reducing larval growth [34], influencing some nervous system and detoxification, and inactivating some hydrolysis enzymes of ACB larvae [35]. Infiltration of DIMBOA into maize leaves stimulated callose accumulation and elevated maize CLA resistance [29]. At 60 h after the 1st instar larvae of *Sesamia nonagrwides* infestation, the leaves of infested maize were injured, with a significant increase in leaf DIMBOA content of 42–96% [36]. Indeed, these studies indicated that bxs metabolites, especially DIMBOA were involved in maize resistance of corn borer and aphid, which can be a good indicator for screening multiple insect-resistant maize genotypes. Results presented here, together with previous findings [29,34,35], showed that both accumulation and catabolism of DIMBOA varied greatly among maize inbred lines, which may contribute to the resistance of borer and aphid in maize (Figure 1A–C and Figure 3C). Meihls et al. [16] also reported that DIMBOA content was highly correlated with its precursor DIMBOA-Glc abundance, and these endogenous compounds potentially built up maize’s resistance to insect pests. Dafoe et al. [37] further found that jasmonic acid and ethylene were produced rapidly in response to corn borer feeding, and their induction differentially regulated bxs in maize stems; even other phytohormones, i.e., salicylic acid and indole-3-acetic acid, generally considered antagonists of jasmonic acid signaling, were also involved in regulating defense responses [38]. 

### 3.2. Genetic Loci Comparison between DIMBOA Content and Multiple Insect-Resistant Traits, and Their Hot Loci Identification

Understanding the genetic basis of the multiple insect resistance of maize is critical to the control of combinatorial attacks of ACB and CLA in the field. In this study, we observed a key trait for multiple insect resistance, i.e., the DIMBOA content in two ecological environments and their BLUP values, to detect nine significant associated SSR markers using association mapping via GLM and MLM across 310 diverse maize inbred lines from Gansu Province, China; these nine SSR markers were located in Bin 1.04, Bin 1.11, Bin 2.01, Bin 4.00, Bin 4.01, Bin 6.02, Bin 8.04, and Bin 10.04, respectively (Figure 4; Table 1). Using the same method, Butrón et al. [15] also identified eight linked single nucleotide polymorphisms (SNP) markers related to DIMBOA content across 281 genetically diverse inbred lines located in Bin 1.04, Bin 2.04, Bin 4.01, Bin 4.04, Bin 5.06, Bin 6.01, Bin 6.05, and Bin 8.06, respectively (Table 2). They also mapped 12 and 7 QTLs associated with DIMBOA/DIMBOA-Glc/DIMBOA-T content in both B73 × CML322 recombinant inbred lines (RILs) and B73 × IL14H RILs populations, respectively; these QTLs were distributed on chromosomes 1, 3, 4, 6, 7, and 8, respectively [15]. These findings demonstrate that there are multiple genetic loci involved in DIMBOA biosynthesis and decomposition on nine chromosomes except for Chromosome 9. Because of their clear differences in genetic effects and phenotypic variance in these loci, there is a great potential to obtain different multiple insect-resistant lines (materials) based on MAS application. In addition, we further combined the genetic loci from previous foliar studies and our current results of DIMBOA content as well as seven other multiple insect-resistant traits (Table 1 and Table 2) to gain a better understanding of hot loci in maize resistance to multiple insect pests and to explore avenues for multiple insect-resistance breeding.

Interestingly, we identified 19 hot loci (Loci 1-Loci 19) involved in maize multiple insect resistance in the present study (Figure 5; Table 3). Of these, Loci 2 in Bin 1.04 (bnlg147-umc1917 interval) is involved in LFR, HO, AR, and DIMBOA content; Loci 4 in Bin 2.00–2.01 (rs624256-umc2363 interval) controls TL and DIMBOA content; Loci 5 in Bin 2.03–2.04 (rs650025-rs65 interval) is responsible for TL and LFR; Loci 8 in Bin 4.00–4.03 (phi072-umc1509 interval) is related to TL, LFR, AR, AIR, AIG, TL/HO, and DIMBOA content; Loci 12 in Bin 5.03 (phi008-umc1935 interval) is associated with AIG and LFR; Loci 13 in Bin 5.05–5.07 (mmc0081-phi128 interval) is associated with LFR, HO, and DIMBOA content; Loci 15 in Bin 8.01–8.03 (umc1139-umc1627 interval) regulates HO and TL; Loci 16 in Bin 8.04–8.05 (umc1858-bnlg1176 interval) is responsible for TL, LFR, and DIMBOA content; Loci 17 in Bin 8.06 (PZA02746-bnlg1031 interval) is related to TL and DIMBOA content; Loci 18 in Bin 9.01 (phi033-umc1958 interval) is involved in TL, HO, and TL/HO; and Loci 19 in Bin 10.04 (umc1336-umc1054 interval) is associated with LFR, HO, and DIMBOA content. Thus, these 11 hot loci have a pleiotropic effect on 2 to 7 multiple insect-resistant traits, and Bin 1.04, Bin 2.00–2.01, Bin 4.00–4.03, Bin 5.05–5.07, Bin 8.04–8.05, Bin 8.06, and Bin 10.04 regions play important roles in conferring DIMBOA accumulation and other aspects of maize multiple-insect resistance to ACB and CLA. Consistent with previous findings, LFR was significantly correlated with HO (phenotypic correlation coefficient (*r*); *r* = 0.252) and TL/HO (*r* = 0.229) in 162 F_3_ maize population to ACB resistance [2]. Meanwhile, 3 of 19 hot loci, i.e., Loci 2, Loci 8, and Loci 12 are co-involved in multiple insect resistance to both ACE and CLA; thus, we speculate that the resistance to ACB and CLA for leaf feeding damage is partially controlled by the same mechanisms in these three hot loci. For future research, the contribution of Loci 2, Loci 8, and Loci 12, as well as their functional genes must be examined when developing elite maize varieties with multiple insect resistance to ACE and CLA.

### 3.3. Validation of Candidate Genes in Hot Loci

According to the physical interval of the above 19 hot loci controlling 8 insect-resistant traits and the GO annotations of corresponding genes in these hot intervals, a total of 49 candidate genes were identified (Table 3 and Appendix A); they may play important roles in maize multiple insect-resistance.

The 12 candidate genes were identified within three hot loci; namely, *GRMZM2G311036* (*bx10*), *GRMZM2G336824* (*bx11*), and *GRMZM2G023325* (*bx12*) were detected in Loci 1 (Bin 1.04), and they encoded DIMBOA-glucoside O-methyltransferase; *GRMZM2G046163* (*lgl*; encoded indole-3-glycerol phosphate lyase) was mapped in Loci 3 (Bin 1.11); *GRMZM2G167549* (*bx3*; encoded indolin-2-one monooxygenase), *GRMZM2G172491* (*bx4*; encoded 3-hydroxy-indolin-2-one monoxygenase (P450)), *GRMZM2G063756* (*bx5*; encoded BHBOA monoxgenase (P450)), *GRMZM6G617209* (*bx6;* encoded 2-oxoglutarate-dependent dioxygenase), *GRMZM2G085054* (*bx8*; encoded 2,4-dihydroxy-7-methoxy-2H-1,4-benzoxazin-3 (4H)-one 2-D-glucosyltransferase), *GRMZM2G085381* (*bx1*; encoded indole-3-glycerol phosphate lyase), and *GRMZM2G085661* (*bx2*; encoded indole-2-monooxygenase) were identified in Loci 8 (Bin 4.00–4.03); and *GRMZM2G441753* (*bx7*; encoded 2,4,7-trihydroxy-2H-1,4-benzoxazin-3(4H)-one (TRIBOA-Glc) O methyl transferase) was located in Loci 9 (Bin 4.04) (Table 3). The *lgl* and *bx1* also displayed similar enzyme functions; *lgl* catalyzed the formation of free indole and was selectively activated by volicitin in the saliva of lepidopterous larvae [15,39,40]. Then, the enzyme actions of four maize cytochrome P450-dependent monooxygenases (*bx2*, *bx3*, *bx4*, and *bx5*) converted free indole to 2,4-dihydroxy-1,4-benoxazin-3-one (DIBOA) [15]. The *bx1* was a modified form of the tryptophan synthase alpha subunit, and it was expressed constitutively in young seedlings, while *igl* was induced in more advanced stages of plant development and contributed to the blend of odors that attracted beneficial parasitoids [15,39,40]. The next step in the bxs biosynthesis pathway was the conversion of DIBOA to DIBOA-Glc by the action of specific gluocosyltransferases. The *bx8* and *GRMZM2G161335* (*bx9*) were involved in the glucosylation of DIBOA [41], and the conversion of DIBOA to DIMBOA-Glc required hydroxylation and methylation. The *bx6* was responsible for the hydroxylation step that converted DIBOA-Glc to TRIBOA-Glc, and this conversion likely took place in the cytosol [42,43]. Methylation was catalyzed by *bx7*, making DIMBOA-Glc [15]. In addition, the *bx10*, *bx11*, and *bx12* were likely candidates for catalyzing the conversion of DIMBOA-Glc to HDMBOA-Glc [16]. These findings also supported the results of the relative expression levels of five candidate genes that controlled DIMBOA biosynthesis; their expression patterns were positively and significantly correlated with the DIMBOA accumulation (Figure 6A–C). Therefore, these candidate genes in Bin 1.04, Bin 1.11, and Bin 4.00–4.03 regions were involved in DIMBOA biosynthesis, and the four associated SSR markers (umc1917, umc1008, umc1017, and umc1758) can be used to distinguish DIMBOA accumulation in maize.

The two candidate genes, *GRMZM2G150256* (*mir2*) and *GRMZM2G150276* (*mir1*) were mapped within Loci 14 and encoded a maize insect resistance-cysteine protease (key defensive protein) against chewing insect pests in maize (Table 3). The synthetic diet aphid feeding trial bioassays with recombinant *mir1*-cysteine protease demonstrated that *mir1*-Cysteine protease triggered direct toxicity to CLA [44], and ethylene acted as a central node in regulating *mir1* expression to different feeding guilds of insect herbivores [44]. 

Other phytohormones may also be responsive to multiple insect-resistance in maize. A total of nine auxin-responsive SAUR family member genes were detected within Loci 4 (Bin 2.00–2.01), Loci 5 (Bin 2.03–2.04), Loci 12 (Bin 5.03), Loci 17 (Bin 8.06), and Loci 18 (Bin 9.01), respectively (Table 3). Similarly, two auxin-responsive SAUR family member genes that closely related to ACB-resistance were detected on chromosome 2 (near rs653464 and rs649775 marker, respectively) based on genome-wide association study (GWAS) [19]. Recent characterization of European corn borer attack on maize stems showed a rapid and sustained accumulation of indole-3-acetic acid and jasmonic acid in damaged tissues [38]. 

Lipoxygenase (LOX) plays critical roles in plant defense against multiple insect pests and pathogens [1,45,46]; and, consistent with LOX’s function, we also found three *LOX* genes, i.e., *GRMZM2G017616* (*LOX9*) within Loci 1 (Bin 1.01–1.02), *GRMZM2G040095* (*LOX6*) within Loci 4 (Bin 2.00–2.01), and *GRMZM2G102760* (*LOX5*) within Loci 12 (Bin 5.03) (Table 3). *LOX6* was strongly induced by jasmonic acid and the fungal pathogen *Cochliobolus carbonum* in maize [47].

The *MYB* transcription factor can interact with mRNA/proteins to form a fine regulatory network to activate the expression of downstream defense genes and induce insect-resistance defense response. Interestingly, the previous findings [19] supported our results, i.e., 12 *MYB* genes were validated within Loci 2 (Bin 1.04), Loci 5 (Bin 2.03–2.04), Loci 10 (Bin 4.05), Loci 13 (Bin 5.05–5.07), Loci 14 (Bin 6.02), Loci 15 (Bin 8.01–8.03), Loci 16 (Bin 8.04–8.05), and Loci 19 (Bin 10.04) in this study, respectively (Table 3). Thus, these MYBs may be involved in the defense response to ACB and CLA, and further studies are needed to explore the downstream target genes of *MYB* in herbivore-induced resistance.

Cell wall invertases (incw) catalyze the irreversible hydrolysis of sucrose into glucose and fructose and play important roles in sucrose partitioning, plant development, and defense responses to biotic stresses [48]. The hydrolysis of cell wall polysaccharides during pathogen infection generates sugar signaling to stimulate intracellular defense response [49]. The five *incw* genes, i.e., *GRMZM2G119941* (*incw4*), *GRMZM2G018716* (*incw7*), and *GRMZM2G018692* (*incw6*) within Loci 4 (Bin 2.00–2.01), *GRMZM2G139300* (*incw1*) within Loci 13 (Bin 5.05–5.07), and *GRMZM2G123633* (*incw3*) within Loci 19 (Bin 10.04) were identified in this study (Table 3). Essmann et al. [50] reported that RNA interference-mediated repression of an *incw* gene resulted in reduced defense of transgenic tobacco plants. Moreover, we also detected *GRMZM2G179679* encoding SWEET 3a (Sugars Will Eventually be Exported Transporter 3a) within Loci 16 (Bin 8.04–8.05) (Table 3). Thereby, sugar metabolism/homeostasis may play a positive role in maize defense response to ACB and CLA.

Plant POD participates in multiple physiological processes, such as auxin metabolism [51], lignin biosynthesis [52,53], and tolerance against osmotic stress [54]. Additionally, reactive oxygen species (ROS), especially H_2_O_2_ induces POD activity, which then oxidizes and polymerizes *p*-coumaryl/coniferyl-/sinapyl-alcohol into lignin monomers on the cell wall [52]; thereby, POD is involved in the loosening and stiffening of the cell wall during plant development. Moreover, López-Castillo et al. [55] reported that *ZmPrx35* as the prevailing POD was involved in defense against pathogens and insects. Similarly, we also identified *Zm00001d024752* (*POD21*) and *Zm00001d052335* (*POD23*) within Loci 19 (Bin 10.04) and Loci 11 (Bin 4.08), respectively (Table 3). Therefore, we speculate that host maize can limit food supplies to multiple insect pests and against larval boring via a key physical barrier, such as cell wall rigidity in the pre-ingestion phase.

In addition, three other candidate genes, i.e., *GRMZM2G102471* (*uce4*) responsible for ubiquitin-conjugating enzyme 4, *RMZM2G130224* responsible for restriction endonuclease type II-like superfamily protein, and *GRMZM2G504910* responsible for tetratricopeptide repeat protein 27 homolog, were identified within Loci 4 (Bin 2.00–2.01), Loci 6 (Bin 2.04), and Loci 7 (Bin 2.05) in the present study, respectively (Table 3). Interestingly, the *GRMZM2G130225* (near rs658849) and *GRMZM2G102471* (near rs624256) were also identified to be involved in TL of ACB using GWAS analysis [19]. 

In summary, according to the above studies, a possible molecular network underlying maize multiple insect-resistance to ACB and CLA was constructed (Figure 7), which could benefit the development of new maize varieties with multiple insect resistance.

### 3.4. Germplasms Diversity and Multiple Insect-Resistance Evaluation

When assessing the genetic diversity in maize genotypes, SSR remains the preferred choice due to their co-dominant and multi-allelic nature, abundance, and loci specificity [23]. In the current study, 748 alleles with a range of 2 to 8 per locus were identified among 310 maize inbred lines using 186 polymorphic SSRs (Appendix A). The finding showed a wide range of diversity among genotypes [23], which will benefit broadening the genetic base in any breeding program. The average PIC value of all SSR markers was 0.543 (range from 0.294 to 0.826) in our result (Appendix A). The data demonstrates the presence of many informative allelic variations in this maize population [22,23]. Moreover, considering the membership probability of ≥ 0.500, the population structure further indicated that the 294 inbred lines were divided into 5 optimal groups (Appendix A); this result was consistent with the findings of Liu et al. [22]. Further, we comprehensively evaluated the multiple insect resistance of 310 inbred lines in 2 ecological environments, and screened 3 high and 15 moderate insect-resistant germplasms, which were mainly in the Reid group (accounting for 50.00%) (Figure 3B–C). These findings have laid a foundation for the improvement of insect-resistant maize varieties in the future.

## 4. Materials and Methods

### 4.1. Plant Materials

In this study, the collected 310 elite maize inbred lines from different ecological environments (Zhangye, 216 lines; Longxi, 64 lines, Jingtai, 21 lines; Pingliang, nine lines) in Gansu Province, China (Appendix A). A total of 20 seeds of these germplasms per row were randomly planted in 4.0 m rows, 0.2 m aisles, and 0.5 m between rows on April 16 in Zhangye (E1; 38.83° N, 106.93° E, 1536 m altitude) and in Longxi (E2; 34.97° N, 104.40° E, 2074 m altitude), Gansu Province, China in 2020. Before sowing, the soil surface was covered with plastic film (0.08 mm thick, 1.2 m wide). At the V6 stage, the 6th leaf of each inbred line in the E1 and E2 environments was collected, frozen in liquid nitrogen, and stored at –70 °C for subsequent DNA extraction and DIMBOA content assay.

### 4.2. DIMBOA Content Assay

The DIMBOA content was determined using high-performance liquid chromatography (HPLC; Shimadzu LCMS8040 system, Beijing, China). Namely, freeze-dried leaves (0.2 g per sample) were homogenized and weighted into screw-capped 10 mL polypropylene centrifuge tubes, and 5 mL of HPLC grade methanol-methanoic acid solution (0.01%, *v*/*v*) was added to each tube. The tubes were rotated and placed in the dark for 12 h and then centrifuged at 12,000 rpm (Centrifuge 5425/5425 R; Eppendorf, Hamburg, Germany) for 20 min at 4 °C. Supernatants (600 μL) were slowly passed the corresponding Millex^®^ needle filter and transferred into auto-sample vials for analysis by HPLC. Standard DIMBOA (CAS No.: 15893-52-4) was purchased (Sigma-Aldrich, MA, USA) and was used to optimize the mass spectrometric parameters and fragment spectra.

### 4.3. Genetic Diversity Analysis and Marker-DIMBOA Content Association Mapping

Genomic DNA was extracted from the 6th leaf of 310 inbred lines using the cetyltrimethyl ammonium bromide (CTAB) method [56]. Then, a total of 186 polymorphic SSR markers covering the entire maize genome from the MaizeGDB (http://www.maizegdb.org/ (accessed on 10 January 2020)) were used to perform SSR analysis. Fragments were separated using polyacrylamide gel electrophoresis. The polymorphism information content (PIC) value and Shannon-Wiener’s index (I) value were determined as follows [23]:(1)$$\mathrm{PIC}=1-\sum^{\;}P_{i}^{2}$$
(2)$$I=-\sum^{\;}P_{i}\mathrm{lnPi}$$
where $P_{i}$ was the i allele frequency. Allele identity was used for an unweighted pair-group method with arithmetic means (UPGMA) cluster analysis. Population structure [57] was analyzed using STRUCTURE v. 2.3.3 software (http://web.stanford.edu/group/pritchardlab/structure_software/release_versions/v2.3.4/html/structure.html (accessed on 10 August 2022)) for the assessment of groups and genetic relationships among the 310 maize inbred lines. The project was run with the set parameters of the population admixture model, and the allele frequency correlated. The optimum group number was determined by ΔK value [58]. A linkage map was developed according to the genetic distance (centimorgan, cM) of corresponding SSR markers on an IBM2 2008 Neighbors map frame (https://www.maizegdb.org/data_center/map (accessed on 18 September 2022)) using BioMercator v. 4.2 software (http://www.bioinformatics.org/mqtl/wiki/ (accessed on 18 September 2022)). The linkage disequilibrium (LD) values for r^2^ [59] and D′ [57] between SSR loci on chromosomes were calculated using Tassel 3.0 software (https://tassel.bitbucket.io/ (accessed on 10 August 2022)), following a permutation test of 10,000. The K matrix and marker-DIMBOA content association mapping was completed in Tassel 3.0 software using the genotypic data of 186 polymorphic SSRs and phenotypic data on DIMBOA content for a set of 310 inbred lines. The association analysis was conducted by a GLM with Q matrix (individuals’ probability of membership in the population) [60] and an MLM with a K + Q matrix [61]. The associated SSRs for DIMBOA content were filtered out based on the phenotypic variance (R^2^) of the marker at *p* < 0.01 level and with the lowest false discovery rate (FDR). Using MaizeGDB (http://www.maizegdb.org/ (accessed on 20 September 2022)) and nucleotide and primer blast tools, the physical locations of associated SSR markers for DIMBOA content were determined on chromosomes.

### 4.4. Hot Genetic Loci and Candidate Genes Detection

Using public databases, i.e., MaizeGDB (http://www.maizegdb.org/ (accessed on 23 September 2022)), NCBI (http://www.ncbi.nlm.nih.gov (accessed on 23 September 2022)), and CNKI (http://www.cnki.net (accessed on 23 September 2022)), we collected information on corresponding QTLs and associated markers for multiple insect-resistant traits from our results and previous studies via QTL analysis, GWAS, and association mapping. The hot genetic loci were the overlapping regions combining multiple genetic loci responsible for multiple insect-resistant traits or single genetic loci that explained the large R^2^ (10%) in 10 Mb physical intervals. Further, the physical map of all hot genetic loci and candidate genes involving multiple insect-resistant traits in these hot genetic loci regions was developed by BioMercator v. 4.2 software (http://www.bioinformatics.org/mqtl/wiki/ (accessed on 28 September 2022)) [62]. The functional annotation of corresponding candidate genes was performed using the tool AgBase v2.00 (https://agbase.arizona.edu/ (accessed on 6 October2022)) [63].

### 4.5. RT-qPCR Analysis

Among 310 maize inbred lines, we selected the two inbred lines with the largest difference in multiple insect resistance, i.e., RX20-1006 with the highest DIMBOA content and T58 with the lowest DIMBOA content. Their total RNAs were extracted from the 6th leaf with TRIZOL reagent (Invitrogen, USA), and cDNA was made using a kit (Starscript II First-stand cDNA Synthesis Mix With gDNA Remover, GenStar, Beijing, China), according to the manufacturer’s instructions. RT-PCR was conducted using TransStart Tip Green qPCR SuperMix (Tran, Beijing, China). The primers for five candidate genes [12,64] were designed using Primer Premier 5.0 software (http://www.premierbiosoft.com/ (accessed on 9 October 2022)) (Appendix A). Relative gene expression levels were assessed by the 2^−∆∆Ct^ method, with *GRMZM2G126010* as an internal reference gene [63].

### 4.6. Statistical Analysis

The average DIMBOA content in 310 inbred lines from five biological replicates in each ecological environment were analyzed, respectively. A mixed linear model was fitted using the lmer function in lme4 package of R (http://www.R-project.org/ (accessed on 16 December 2021)) to calculate the BLUP of DIMBOA content values [65]. These data were then compared statistically using IBM-SPSS Statistics v. 19.0 (SPSS Inc., Chicago, IL, USA; http://www.ibm.com/products/spss-statistics (accessed on 16 December 2021)). The significance of the total and residual variances of DIMBOA content in 310 inbred lines under both ecological environments was estimated by a GLM for univariate data and by one-way analysis of variance (ANOVA). The broad-sense heritability ($H_{B}^{2}$) and genotype × environment interaction heritability ($H_{GE}^{2}$) values for DIMBOA content under both environments were estimated as follows [53,66,67]:*H_B_*^2^ = σ_g_^2^/(σ_g_^2^ + σ_ge_^2^/n + σ_ε_^2^/nr),(3)
*H_GE_^2^* = (σ_g_^2^/n)/(σ_g_^2^ + σ_ge_^2^/n + σ_ε_^2^/nr)(4)
where σ_g_^2^ was the genotypic variance, σ_e_^2^ was the environmental variance, σ_ε_^2^ was the error variance, σ_ge_^2^ was the variance of genotype × environment interaction, n was the number of ecological environments (n = 2), and r was the number of replications (r = 5). The CVg [63] of DIMBOA content among all inbred lines under each environment was calculated as follows:(5)$${CV}_{g}\;=\;\delta /\bar{X}\times 100\%$$
where $\bar{X}$ was the average value of DIMBOA content among all inbred lines in each environment, and δ was the standard deviation.

## 5. Conclusions

In summary, the DIMBOA confers significant resistance to ACB and CLA. In this study, SSR analysis revealed a wide genetic diversity in the 310 tested maize inbred lines from four type regions of China’s Gansu Province, which is the largest maize seed production and breeding area in China. Population structure indicated that 294 inbred lines were successfully assigned to one or another group at a membership probability of ≥ 0.500. DIMBOA performance evaluation screened out 3 high and 15 moderate insect-resistant inbred lines, which can be used as parents in breeding programs to develop new maize varieties with multiple insect resistance. Using linkage mapping, we detected nine significant SSRs associated with DIMBOA content in both environments. We then combined the 47 original genetic loci for 8 multiple insect-resistant traits from previous studies to detect 19 hot loci. Among them, 11 hot loci were located in Bin 1.04, Bin 2.00–2.01, Bin 2.03–2.04, Bin 4.00–4.03, Bin 5.03, Bin 5.05–5.07, Bin 8.01–8.03, Bin 8.04–8.05, Bin 8.06, Bin 9.01, and Bin 10.04 regions, and they supported pleiotropy for their association with two or more insect-resistant traits. Further, the 49 candidate genes involved in DIMBOA biosynthesis, sugar metabolism/homeostasis, and other multiple insect-resistant defense mechanisms in maize were identified in all 19 hot loci, and their highly interconnected network may form complex maize, multiple insect, pest-induced defense mechanisms.

## Figures and Tables

**Figure 1 ijms-24-02138-f001:**
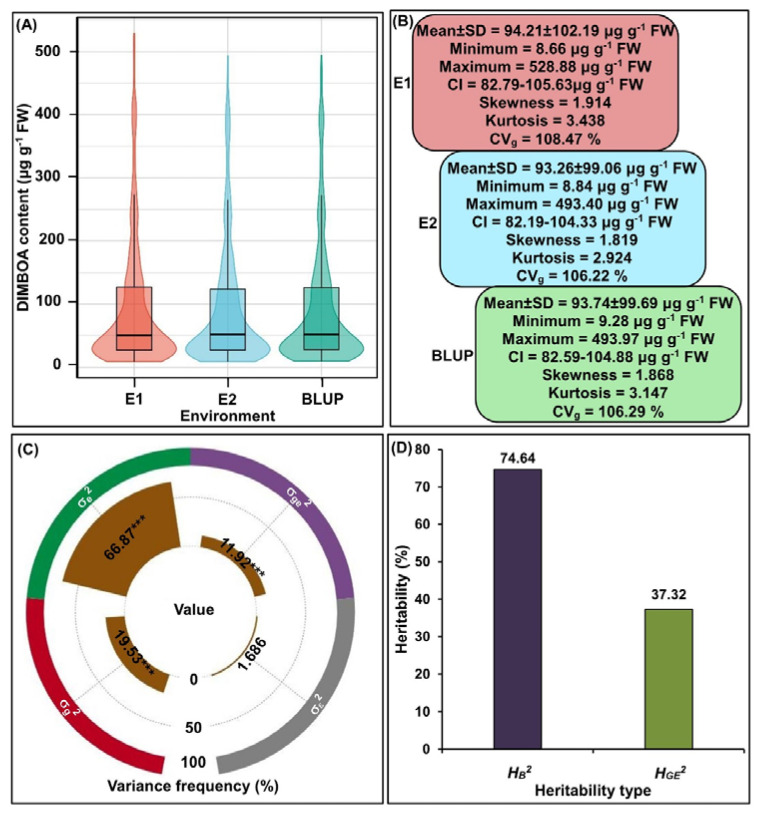
Statistical analysis of DIMBOA content at V6 stage in Zhangye (E1) and Longxi (E2) ecological environments among 310 maize inbred lines seedlings in Gansu Province, China in 2020 and their best linear unbiased prediction (BLUP) values. (**A**) Violin plot shows the DIMBOA content in E1 and E2 environments, and their BLUP values among all inbred lines. (**B**) The statistics including mean ± standard deviation (SD), minimum, maximum, confidence interval (CI), skewness, kurtosis, and genetic variation coefficient (CVg) of DIMBOA content and their BLUP values among all inbred lines in E1 and E2 environments. (**C**) Frequency of the genotypic variance (σ_g_^2^), environmental variance (σ_e_^2^), genotype × environment interaction variance (σ_ge_^2^), and error variance (σ_ε_^2^) for DIMBOA content; *** indicated a significant difference with *p* < 0.01 (ANOVA). (**D**) The broad-sense heritability ($H_{B}^{2}$) and genotype × environment interaction heritability ($H_{GE}^{2}$) of DIMBOA content.

**Figure 2 ijms-24-02138-f002:**
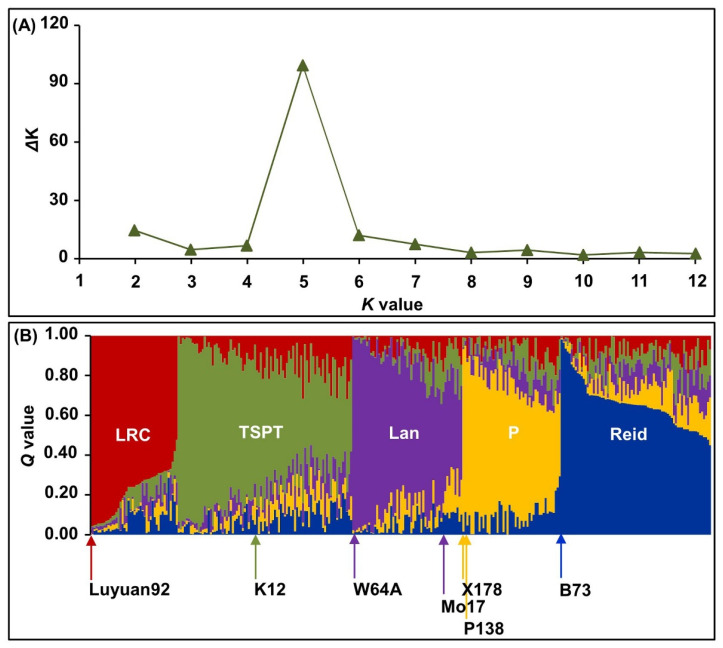
Population structure of 310 maize inbred lines revealed by 186 polymorphic SSR markers. (**A**) Change curve in the log probability data of ΔK value against K (group number) value. (**B**) Population structure of 310 inbred lines based on 186 SSR markers at K = 5. Each inbred line was represented by a vertical line, which indicated the membership coefficient for that individual, and the five groups were the Lüda red cob (LRC), Tang si ping tou (TSPT), Lancaster (Lan), P, and Reid groups, respectively.

**Figure 3 ijms-24-02138-f003:**
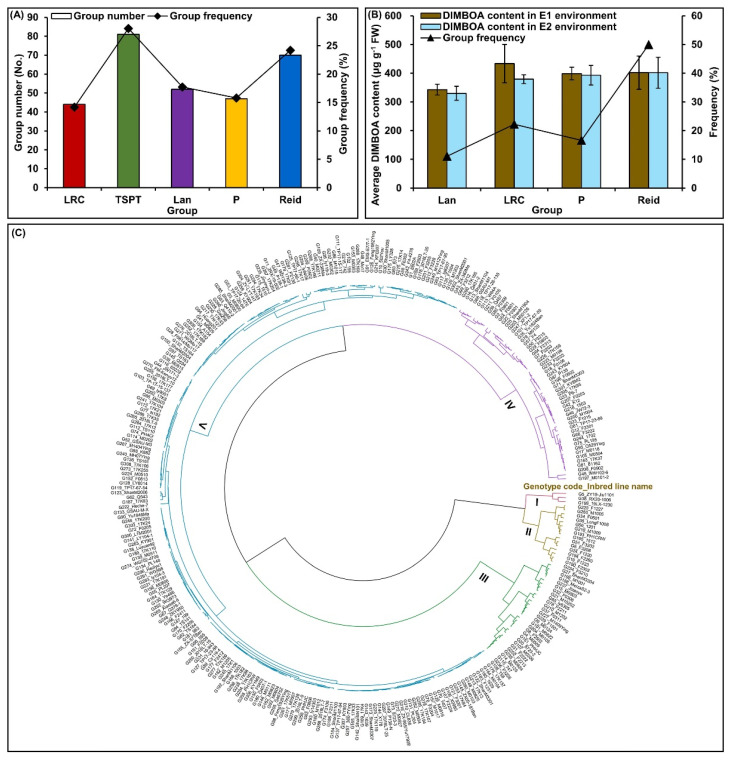
Evaluation of multiple insect-resistance among 310 maize inbred lines in both Zhangye (E1) and Longxi (E2) ecological environments by unweighted pair-group method with arithmetic means (UPGMA) cluster analysis, and their attributive group dissection. (**A**) Statistics for group number and frequency of each optimal group. (**B**) Attributive groups and their frequency distribution of three high insect-resistant and 15 moderate insect-resistant inbred lines; average of DIMBOA content for each attributive group (including Lüda red cob (LRC), Lancaster (Lan), P, and Reid group, respectively) in E1 and E2, respectively. (**C**) Evaluation of multiple insect-resistance among 310 inbred lines in two environments were performed using R package (http://www.R-project.org/; accessed on 20 April 2022) with UPGMA (the K value sets to 5), including type I (high insect-resistant lines), type II (moderate insect-resistant lines), type III (insect-resistant lines), type IV (moderate insect-susceptible lines), and type V (insect-susceptible lines), respectively.

**Figure 4 ijms-24-02138-f004:**
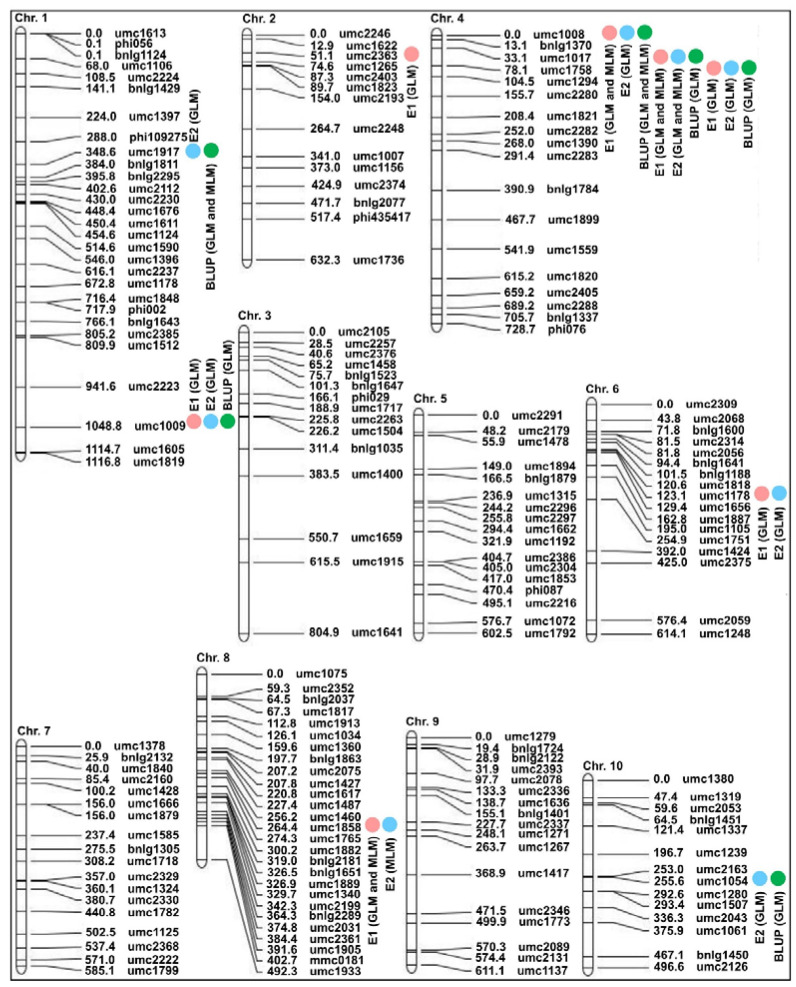
Genetic linkage map construction according to the genetic distance (centimorgan, cM) of 186 polymorphic SSRs on IBM2 2008 Neighbors map frame. The identified SSRs were significantly (*p* < 0.01) associated with DIMBOA content in Zhangye (E1; pink circle) and Longxi (E2; blue circle) ecological environments, Gansu Province, China in 2020. The best linear unbiased prediction (BLUP; green circle) values for these SSRs were assessed using a general linear model (GLM) and mixed linear model (MLM), respectively.

**Figure 5 ijms-24-02138-f005:**
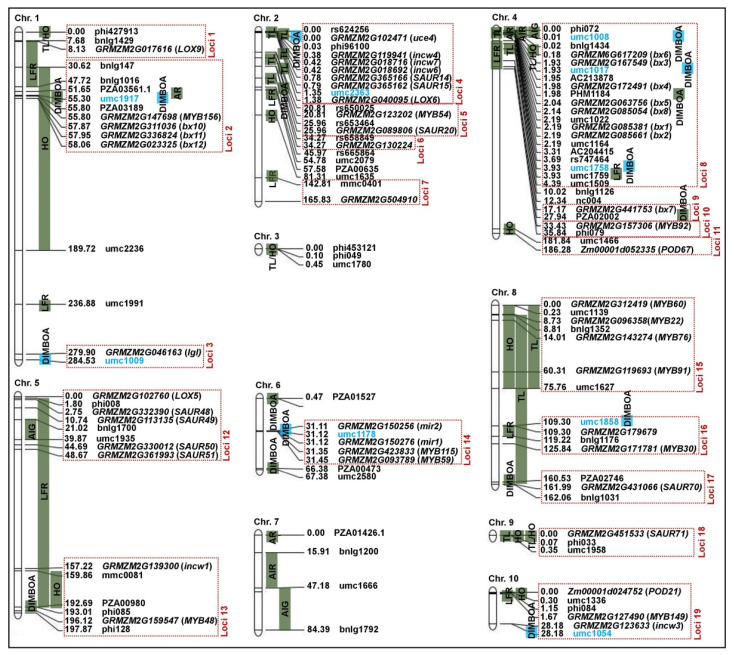
Physical map construction, and the distribution of hot genetic loci (Loci, red dashed box), candidate genes, and original genetic loci for multiple insect-resistant traits including tunnel length of corn borer (TL), aphid incidence rate (AIR), average aphid incidence grade (AIG), aphid resistance (AR), DIMBOA content, leaf feeding rating of corn borer (LFR), number of holes of corn borer (HO), and tunnel length/number of holes of corn borer (TL/HO) in the physical map. The light green rectangles represent original genetic loci from previous studies. The blue rectangles represent our results. The blue markers represent the associated markers in our results.

**Figure 6 ijms-24-02138-f006:**
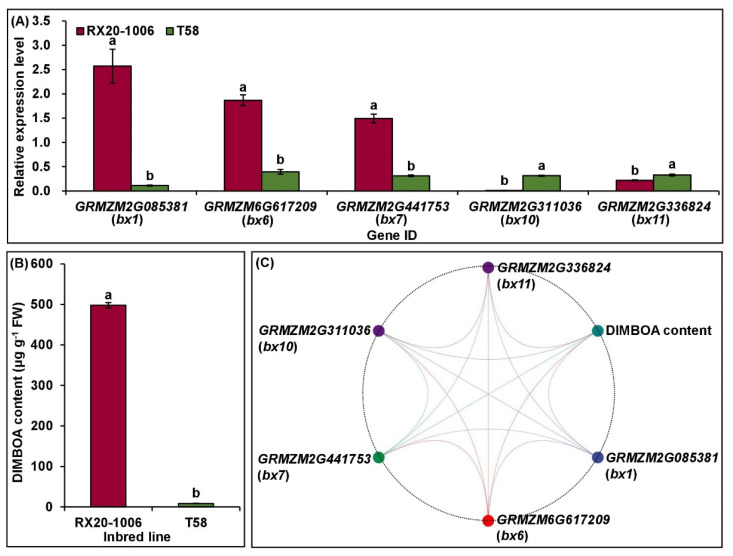
The relative expression level of five candidate genes involved in DIMBOA biosynthesis, DIMBOA content observation and their correlation in maize seedlings of RX20-1006 (high insect-resistant line) and T58 (insect-susceptible line) in Zhangye (E1) environment, Gansu Province, China. (**A**) The relative expression level (mean ± standard deviation) of the five candidate genes; different lowercase letters indicate a significant difference (*p* < 0.05) in gene expression. (**B**) DIMBOA content (mean ± standard deviation) in high insect-resistant (RX20-1006) and insect-susceptible line (T58); different lowercase letters indicate a significance difference (*p* < 0.05) in DIMBOA content. (**C**) The correlational relationships between the relative expression level of five candidate genes and DIMBOA content; lines represent significant correlations (*p* < 0.05).

**Figure 7 ijms-24-02138-f007:**
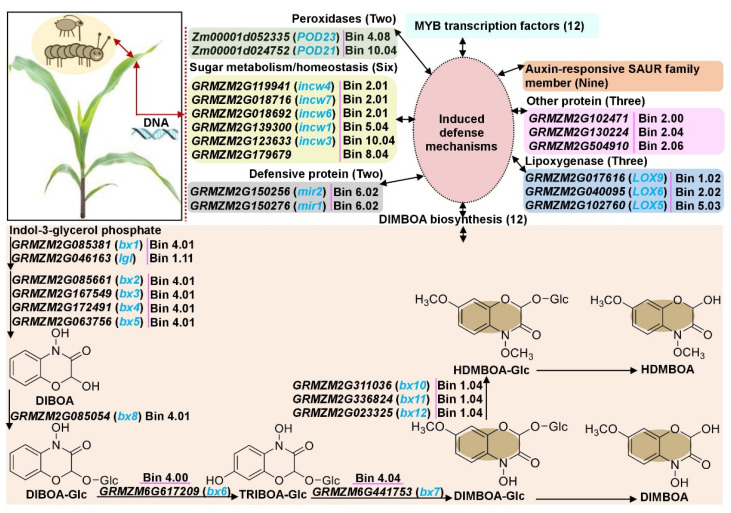
Molecular defense mechanisms underlying resistance to multiple insect pests of Asian corn borer (ACB) and corn leaf aphid (CLA) in maize. The synergy and antagonism of 49 candidate genes within 19 hot genetic loci formed the complex maize multiple insect pest-induced defense mechanisms, as well as DIMBOA biosynthesis pathway network.

**Table 1 ijms-24-02138-t001:** SSR markers were found significantly (*p* < 0.01) associated with DIMBOA content in both ecological environments, and their best linear unbiased prediction (BLUP) values in marker-trait association analysis were assessed using a general linear model (GLM) and mixed linear model (MLM), respectively.

Code	Associated SSR Marker	Bin Location	Contig	Physical Location (bp)	GLM			MLM		
					*R*^2^ in E1	*R*^2^ in E2	*R*^2^ in BLUP	*R*^2^ in E1	*R*^2^ in E2	*R*^2^ in BLUP
1	umc1917	1.04	ctg14	62,677,623 to 62,680,874		6.14	11.44			9.29
2	umc1009	1.11	ctg64	292,965,682 to 292,971,925	13.52	10.16	10.97			
3	umc2363	2.01	ctg69	4,164,953 to .4,170,212	4.30					
4	umc1008	4.00	ctg154	1,078,106 to 1,080,829	15.39	20.04	12.61	9.48		13.70
5	umc1017	4.01	ctg155	3,002,162 to 3,006,932	19.11	10.83	19.42	10.57	9.94	
6	umc1758	4.01	ctg156	5,004,917 to 5,009,075	10.53	16.74	9.90			
7	umc1178	6.02	ctg281	89,065,425 to 89,072,192	7.97	9.19				
8	umc1858	8.04	ctg349	112,055,118 to ..112,059,831	5.07			4.92	4.74	
9	umc1054	10.04	ctg412	114,307,530 to 114,311,922		7.59	5.41			

SSR, simple sequence repeat; GLM, general linear model; MLM, mixed linear model; BLUP, best linear unbiased prediction; *R*^2^ in E1/E2, the phenotypic variation explained by associated SSR loci for DIMBOA content in Zhangye/Longxi, Gansu Province, China in 2020; *R*^2^ in BLUP, the phenotypic variation explained by associated SSR loci for BLUP values.

**Table 2 ijms-24-02138-t002:** Summary of original genetic loci for multiple maize insect-resistant traits of Asian corn borer (ACB) and corn leaf aphid (CLA) from previous studies.

Trait	Marker Interval	Chr. (Bin Location)	Marker Type	Marker Physical Location (bp)	Contig	*R^2^* (%)	Population (Size)	Method	Reference
TL	rs747464	4 (4.01)	SNP	4,759,749			Inbred lines (301)	GWAS	Gao 2018 [19]
	rs665864	2 (2.04)	SNP	48,777,177					
	rs624256	2 (2.00)	SNP	2,806,677 to 2,812,518					
	rs653464	2 (2.04)	SNP	28,768,661 to 28,772,516					
	rs650025	2 (2.03)	SNP	23,620,411 to 23,625,163					
	rs658849	2 (2.04)	SNP	37,078,026 to .37,083,625					
AIR	phi072–umc1164	4 (4.00–4.01)	SSR	1,078,106 to 1,080,829 3,264,213 to 3,268,368	ctg154 ctg155	9.26/ 11.55	BT-1 × N6 RILs (250)	QTL	Li 2016 [24]
	umc1666–bnlg1200	7 (7.01–7.02)	SSR	47,948,200 to 47,950,815 16,678,518 to 16,683,850	ctg301 ctg297	6.54			
AIG	phi072–umc1164	4 (4.00–4.01)	SSR	1,078,106 to 1,080,829 3,264,213 to 3,268,368	ctg154 ctg155	8.60/ 11.53			
	bnlg1700–umc1935	5 (5.03–5.03)	SSR	33,303,797 to 33,305,797 52,145,125 to 52,169,752	ctg217 ctg219	4.73			
	bnlg1792–umc1666	7 (7.02–7.02)	SSR	85,162,973 to 85,165,402 47,948,200 to 47,950,815	ctg305 ctg301	6.59			
AR	PZA03561.1	1 (1.04)	SNP	60,091,612 to 60,097,741			B73 × Ky21 RILs (122)	QTL	Tzin et al. 2015 [12]
	PZA01426.1	7 (7.00)	SNP	774,537 to 776,537					
AR	AC213878–AC204415	4 (4.01–4.01)	SNP	3,021,364 to 3,062,132 4,379,100 to 4,395,483	ctg155 ctg156		B73 × Mo17 RILs (142)	QTL	Betsiashvili et al. 2015 [4]
DIMBOA	PZA03189	1 (1.04)	SNP	64,242,765 to 64,246,594	ctg14	2.80	Genetically diverse inbred lines (281)	GLM	Butrón et al. 2010 [15]
	PZA00635	2 (2.04)	SNP	60,393,632 to 60,395,632	ctg80	3.41			
	PHM1184	4 (4.01)	SNP	3,050,215 to 3,055,036	ctg155	15.74			
	PZA02002	4 (4.04)	SNP	29,007,117 to 29,016,300	ctg163	2.44			
	PZA00980	5 (5.06)	SNP	204,965,179 to 204,967,179		3.34			
	PZA01527	6 (6.01)	SNP	58,422,170 to 58,431,307		2.40			
	PZA00473	6 (6.05)	SNP	124,327,526 to 124,335,707		1.61			
	PZA02746	8 (8.06)	SNP	163,286,695 to 163,294,127	ctg362	2.26			
LFR	umc1991	1 (1.08)	SSR	245,317,331 to 245,319,777	ctg50	7.24	Mc37 × Zi330 F_2:3_ (162)	QTL	Li et al. 2010 [2]
	umc2079	2 (2.04)	SSR	57,593,899 to 57,596,593	ctg79	7.01			
	mmc0401	2 (2.05)	SSR	145,615,676 to 145,621,417	ctg90	19.28			
	umc1759	4 (4.01)	SSR	5,004,917 to 5,009,075	ctg156	12.18			
	phi084	10 (10.04)	SSR	87,282,919 to 87,285,844	ctg406	15.27			
HO	umc1635	2 (2.05)	SSR	84,120,011 to 84,124,101	ctg85	12.63			
	phi033	9 (9.01)	SSR	11,499,426 to 11,509,187	ctg371	6.03			
TL/HO	phi427913	1 (1.01)	SSR	8,441,182 to 8,447,730	ctg4	13.01			
	umc1958	9 (9.01)	SSR	11,778,648 to 11,782,406	ctg371	6.54			
LFR	bnlg1429–bnlg1016	1 (1.02–1.04)	SSR	16,117,043 to 16,123,450 56,158,913 to 56,164,047	ctg7 ctg14	12.80	H21 × Mo17 F_2:3_ (120)	QTL	Yu 2003 [20]
	phi008–phi085	5 (5.03–5.06)	SSR	14,083,193 to 14,086,595 205,292,914 to 205,298,428	ctg209 ctg251	12.80			
	umc1858–bnlg1176	8 (8.04–8.05)	SSR	112,055,118 to 112,059,831 121,984,323 to 121,999,950	ctg349 ctg4	35.10			
HO	umc1466	4 (4.08)	SSR	182,908,212 to 182,910,212	ctg184	50.80			
	umc1336	10 (10.04)	SSR	86,430,405 to 86,433,004	ctg406	51.80			
TL	bnlg1434–bnlg1126	4 (4.01–4.03)	SSR	1,093,958 to 1,099,174 11,086,040 to 11,089,955	ctg154 ctg184	7.70			
	bnlg1352–bnlg1031	8 (8.02–8.06)	SSR	11,570,168 to 11,575,230 164,821,348 to 164,827,624	ctg326 ctg363	11.00			
TL/HO	umc1022–nc004	4 (4.01–4.03)	SSR	3,259,762 to 3,264,006 13,408,584 to 13,413,986	ctg155 ctg158	12.70			
LFR	umc1509–phi079	4 (4.02–4.05)	SSR	5,459,368 to 5,463,263 36,911,994 to 36,918,779	ctg156 ctg164	15.80	Zi330 × K36 F_2:3_ (114)	QTL	Yu 2003 [20]
HO	bnlg147–umc2236	1 (1.02–1.06)	SSR	39,062,881 to 39,072,502 198,164,070 to 198,166,779	ctg11 ctg41	9.60			
	mmc0081–phi128	5 (5.05–5.07)	SSR	172,138,316 to 172,143,312 210,153,373 to 210,156,706	ctg238 ctg253	12.10			
	umc1139–umc1627	8 (8.01–8.03)	SSR	2,989,632 to 3,048,953 78,519,415 to 78,523,602	ctg326 ctg344	15.90			
TL	phi96100	2 (2.01)	SSR	2,835,084 to 2,837,748	ctg68	49.60			
	umc1139–umc1627	8 (8.01–8.03)	SSR	2,989,632 to 3,048,953 78,519,415 to 78,523,602	ctg326 ctg344	15.80			
	phi033–umc1958	9 (9.01–9.01)	SSR	11,237,266 to 11,239,266 11,778,648 to 11,782,406	ctg371 ctg371	12.80			
TL/HO	phi049–phi453121	3 (3.00–3.01)	SSR	1,728,270 to 1,730,270 1,627,384 to 1,635,027	ctg111 ctg114	8.80			

TL, tunnel length of corn borer; AIR, aphid incidence rate; AIG, average aphid incidence grade; AR, aphid resistance; DIMBOA, DIMBOA content; LFR, leaf feeding rating of corn borer; HO, number of holes of corn borer; TL/HO, tunnel length/number of holes of corn borer; Chr., chromosome; SNP, single nucleotide polymorphisms; SSR, simple sequence repeat; *R*^2^, the phenotypic variation explained by identified genetic loci; RILs, recombinant inbred lines; GWAS, genome-wide association studies; QTL, quantitative trait loci; GLM, general linear model.

**Table 3 ijms-24-02138-t003:** The hot genetic loci of multiple insect-resistance and corresponding candidate genes in these hot genetic loci.

Hot Loci	Bin Location	Candidate Gene ID	Encoded Protein	Gene Location (bp)	Orthologs
Loci 1	1.01–1.02	GRMZM2G017616 (LOX9)	Lipoxygenase 9	Bin 1.02 16,572,327 to 16,582,222	LOC_Os03g08220 (Oryza sativa)
Loci 2	1.04	GRMZM2G147698 (MYB156)	MYB-transcription factor 156	Bin 1.04 64,242,265 to 64,247,094	LOC_Os03g25550 (Oryza sativa)
		GRMZM2G311036 (bx10)	DIMBOA-glucoside O-methyltransferase	Bin 1.04 66,309,137 to 66,314,243	Sb01g033880 (Sorghum bicolor)
		GRMZM2G336824 (bx11)	DIMBOA-glucoside O-methyltransferase	Bin 1.04 66,391,909 to 66,397,349	Sb01g033880 (Sorghum bicolor)
		GRMZM2G023325 (bx12)	DIMBOA-glucoside O-methyltransferase	Bin 1.04 66,504,957 to 66,508,971	Sb01g033880 (Sorghum bicolor)
Loci 3	1.11	GRMZM2G046163 (lgl)	Indole-3-glycerol phosphate lyase	Bin 1.11 288,336,957 to 288,342,082	LOC_Os03g58290 (Oryza sativa)
Loci 4	2.00–2.01	GRMZM2G102471 (uce4)	Ubiquitin-conjugating enzyme 4	Bin 2.00 2,806,677 to 2,812,518	LOC_Os04g57220 (Oryza sativa)
		GRMZM2G119941 (incw4)	Invertase cell wall 4	Bin 2.01 3,188,791 to 3,195,146	LOC_Os04g56920 (Oryza sativa)
		GRMZM2G018716 (incw7)	Invertase cell wall 7	Bin 2.01 3,227,856 to 3,233,490	LOC_Os04g56930 (Oryza sativa)
		GRMZM2G018692 (incw6)	Invertase cell wall 6	Bin 2.01 3,232,233 to 3,237,644	LOC_Os04g56920 (Oryza sativa)
		GRMZM2G365166 (SAUR14)	Auxin-responsive protein SAUR36	Bin 2.01 3,590,872 to 3,594,536	LOC_Os04g56690 (Oryza sativa)
		GRMZM2G365162 (SAUR15)	Indole-3-acetic acid-induced protein ARG7	Bin 2.01 3,600,982 to 3,605,199	LOC_Os04g56680 (Oryza sativa)
		GRMZM2G040095 (LOX6)	Lipoxygenase 6	Bin 2.02 4,190,652 to 4,197,763	Sb06g031350 (Sorghum bicolor)
Loci 5	2.03–2.04	GRMZM2G123202 (MYB54)	MYB-transcription factor 54	Bin 2.03 23,620,411 to 23,625,163	LOC_Os04g45060 (Oryza sativa)
		GRMZM2G089806 (SAUR20)	Auxin-induced protein X10A	Bin 2.04 28,768,661 to 28,772,516	LOC_Os04g43740 (Oryza sativa)
Loci 6	2.04	GRMZM2G130224	Restriction endonuclease type II-like superfamily protein	Bin 2.04 37,078,026 to 37,083,625	LOC_Os04g40900 (Oryza sativa)
Loci 7	2.05	GRMZM2G504910	Tetratricopeptide repeat protein 27 homolog	Bin 2.06 168,638,340 to 168,652,992	LOC_Os07g27180 (Oryza sativa)
Loci 8	*4.00* *–4.03*	*GRMZM6G617209 (bx6)*	2-oxoglutarate-dependent dioxygenase	Bin 4.00 1,251,138 to 1,255,544	LOC_Os03g48430.1 (Oryza sativa)
		GRMZM2G167549 (bx3)	Indolin-2-one monooxygenase	Bin 4.01 3,001,662 to 3,007,432	Sb05g026080 (Sorghum bicolor)
		GRMZM2G172491 (bx4)	3-hydroxy-indolin-2-one monoxygenase (P450)	Bin 4.01 3,049,715 to 3,055,536	Sb05g026080 (Sorghum bicolor)
		GRMZM2G063756 (bx5)	BHBOA monoxgenase (P450)	Bin 4.01 3,111,425 to 3,117,001	LOC_Os08g01510.1 (Oryza sativa)
		GRMZM2G085054 (bx8)	2,4-dihydroxy-7-methoxy-2H-1,4-benzoxazin-3 (4H)-one 2-D-glucosyltransferase	Bin 4.01 3,213,147 to 3,218,058	LOC_Os11g25454.1 (Oryza sativa)
		GRMZM2G085381 (bx1)	Indole-3-glycerol phosphate lyase	Bin 4.01 3,259,262 to 3,264,506	LOC_Os03g58300.1 (Oryza sativa)
		GRMZM2G085661 (bx2)	Indole-2-monooxygenase	Bin 4.01 3,263,713 to 3,268,868	Sb05g026080 (Sorghum bicolor)
Loci 9	4.04	GRMZM2G441753 (bx7)	TRIBOA-glc O methyl transferase	Bin 4.04 18,243,663 to 18,246,553	LOC_Os12g25450.1 (Oryza sativa)
Loci 10	*4.05*	*GRMZM2G157306 (MYB92)*	MYB-related-transcription factor 92	Bin 4.05 34,497,413 to 34,505,954	LOC_Os08g05510 (Oryza sativa)
Loci 11	4.08	Zm00001d052335 (POD23)	Peroxidase 23	Bin 4.08 187,346,362 to 187,351,257	
Loci 12	5.03	GRMZM2G102760 (LOX5)	Lipoxygenase 5	Bin 5.03 12,284,156 to 12,292,064	LOC_Os03g49380 (Oryza sativa)
		GRMZM2G332390 (SAUR48)	Auxin-responsive SAUR family member	Bin 5.03 15,029,401 to 15,033,175	LOC_Os03g45850 (Oryza sativa)
		GRMZM2G113135 (SAUR49)	Auxin-responsive SAUR family member	Bin 5.03 60,953,699 to 60,958,116	LOC_Os10g36703 (Oryza sativa)
		GRMZM2G330012 (SAUR50)	Auxin-responsive protein SAUR19	Bin 5.03 56,965,982 to 56,969,995	LOC_Os06g48860 (Oryza sativa)
		GRMZM2G361993 (SAUR51)	Auxin-responsive protein SAUR32	Bin 5.03 60,953,699 to 60,958,116	LOC_Os06g50040 (Oryza sativa)
Loci 13	5.05–5.07	GRMZM2G139300 (incw1)	Invertase cell wall 1	Bin 5.04 169,496,097 to 169,503,589	LOC_Os02g33110 (Oryza sativa)
		GRMZM2G159547 (MYB48)	MYB-transcription factor 48	Bin 5.07 208,396,628 to 208,401,248	LOC_Os02g51799 (Oryza sativa)
Loci 14	6.02	GRMZM2G150256 (mir2)	Maize insect resistance 2-cysteine protease	Bin 6.02 89,064,767 to 89,072,691	Sb10g028000 (Sorghum bicolor)
		GRMZM2G150276 (mir1)	Maize insect resistance 1-cysteine protease	Bin 6.02 89,070,747 to 89,075,683	Sb10g028010 (Sorghum bicolor)
		GRMZM2G423833 (MYB115)	MYB-transcription factor 115	Bin 6.02 89,299,116 to 89,305,100	LOC_Os06g40330.1 (Oryza sativa)
		GRMZM2G093789 (MYB59)	MYB-transcription factor 59	Bin 6.02 89,401,621 to 89,417,741	LOC_Os06g40330.1 (Oryza sativa)
Loci 15	8.01–8.03	GRMZM2G312419 (MYB60)	MYB-transcription factor 60	Bin 8.01 2,760,493 to 2,764,284	LOC_Os01g16810 (Oryza sativa)
		GRMZM2G096358 (MYB22)	MYB-transcription factor 22	Bin 8.02 11,485,302 to 11,489,135	LOC_Os01g03720 (Oryza sativa)
		GRMZM2G143274 (MYB76)	MYB-transcription factor 76	Bin 8.02 16,771,384 to 16,774,583	LOC_Os01g07430 (Oryza sativa)
		GRMZM2G119693 (MYB91)	MYB-transcription factor 91	Bin 8.03 63,067,162 to 63,070,647	LOC_Os05g46610 (Oryza sativa)
Loci 16	8.04–8.05	GRMZM2G179679	Sugars will eventually be exported transporter 3a	Bin 8.04 112,055,118 to 112,059,831	LOC_Os05g12320 (Oryza sativa)
		GRMZM2G171781 (MYB30)	MYB-transcription factor 30	Bin 8.05 128,604,416 to 128,609,501	LOC_Os05g04820 (Oryza sativa)
Loci 17	8.06	GRMZM2G431066 (SAUR70)	Auxin-responsive protein SAUR50	Bin 8.06 164,748,431 to 164,752,132	
Loci 18	9.01	GRMZM2G451533 (SAUR71)	SAUR-like auxin-responsive protein family	Bin 9.01 11,435,786 to 11,439,677	LOC_Os06g09550 (Oryza sativa)
Loci 19	10.04	Zm00001d024752 (POD21)	Peroxidase 21	Bin 10.04 86,134,891 to 86,138,913	
		GRMZM2G127490 (MYB149)	MYB-transcription factor 149	Bin 10.04 87,799,992 to 87,806,148	LOC_Os05g04820 (Oryza sativa)
		GRMZM2G123633 (incw3)	Invertase cell wall 3	Bin 10.04 114,307,030 to 114,312,422	LOC_Os04g33720 (Oryza sativa)

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
