# Peer review of "Genetic Variation, DIMBOA Accumulation, and Candidate Gene Identification in Maize Multiple Insect-Resistance"

_ijms, 2023, doi:10.3390/ijms24032138_

Round 1
Reviewer 1 Report
Maize, which is utilized globally as 35 food, animal feed, and biofuel products, is one of the most important agro-economical crops. Despite the large production area for maize, extensive losses are common in China due to a range of biotic stressors, such as multiple insect injury.
At present, although numerous genetic loci for multiple insect-resistant traits have been identified, little is known about genetic controls regarding DIMBOA content. In the present study, the best linear unbiased prediction (BLUP) values of DIMBOA content in two ecological environments across 310 maize inbred lines were calculated, and their phenotypic data 15 and BLUP values were used for marker-trait association analysis. The identified insect-resistant inbred lines, markers associated with DIMBOA content, and other multiple insect-resistant traits to ACB and CLA, as well as the corresponding candidate genes may be validated and utilized in MAS for the development of suitable varieties.
The finding by itself is highly interesting. Overall, the present manuscript is well written, with the sound experimental designs and scientific data analysis.
From all the above, I recommend that it can be accepted for publication in its present form.
Author Response
Dear Editor and Reviewers
Thank you for your letter of – and for the referee’s comments concerning our manuscript, “Genetic Variation, DIMBOA Accumulation, and Candidate Genes Identification in Maize Multiple Insect-Resistance (ijms-2121902)”. We have carefully studied these comments and have made corresponding corrections to the manuscript, which we describe in detail below. We would like to re-submit the manuscript and that for possible publication on the Special Issue: “Molecular Research in Maize” of International Journal of Molecular Sciences. Thank you very much for your time and consideration.
Editor:
Thank you again for your manuscript submission. As the last referee has not finalized his/her review, we are now sending the available comments to you for revision to save more time. Once we receive the last report, we will send it to you immediately. We will also keep you informed if we decide to cancel the review request:
Thanks for the positive comments of you and all reviewers for our manuscript. As suggested, we have carefully revised and improved the manuscript. We then have re-submitted the manuscript.
Thank you for your consideration.
Your manuscript has been reviewed by experts in the field. Please find your manuscript with the referee reports at this link: https://susy.mdpi.com/user/manuscripts/resubmit/0b92bf215f4e223c5f4066 efe9bd3e15.
Thanks for the positive comments of you and all reviewers for our manuscript. As suggested, we have carefully revised and improved the manuscript using the “Track Changes” function of our manuscript at the above link. We then have re-submitted the manuscript within the allotted time.
Thank you for your consideration.
(I) Please revise your manuscript according to the referees’ comments and upload the revised file within 5 days.
Thanks for your positive comments for our manuscript. As suggested, we have carefully revised the manuscript. We then have re-submitted the manuscript within the allotted time.
Thank you for your consideration.
(II) Please use the version of your manuscript found at the above link for your revisions.
Thanks for the positive comments for our manuscript. As suggested, we have carefully revised and improved the manuscript using the versions of our manuscript at the above link. We then have re-submitted the manuscript.
Thank you for your consideration.
(III) Please check that all references are relevant to the contents of the manuscript.
Thanks for your positive comments for our manuscript. As suggested, we have carefully checked all references to make sure they are relevant to the contents of the manuscript. We then have re-submitted the manuscript.
Thank you for your consideration.
(IV) Any revisions made to the manuscript should be marked up using the “Track Changes” function if you are using MS Word/LaTeX, such that changes can be easily viewed by the editors and reviewers.
Thanks for your positive comments for our manuscript. As suggested, we have carefully revised and improved the manuscript using the “Track Changes” function of our manuscript at the above link. We then have re-submitted the manuscript within the allotted time.
Thank you for your consideration.
(V) Please provide a short cover letter detailing your changes for the editors’ and referees’ approval.
Thanks for your positive comments for our manuscript. As suggested, we have carefully revised and improved the manuscript. In addition, we have prepared a detailed response letter to all reviewers for each point, and then have re-submitted the manuscript.
Thank you for your consideration.
If one of the referees has suggested that your manuscript should undergo extensive English revisions, please address this issue during revision. We propose that you use one of the editing services listed at https://www.mdpi.com/authors/english or have your manuscript checked by a native English-speaking colleague.
Thanks for the positive comments you and all reviewers for our manuscript. The English language of the manuscript has been well modified by Charlesworth Author Services (https://www.cwauthors.com.cn/), and we provide the proof of English language polish. We then have re-submitted the manuscript.
Thank you for your consideration.
Reviewer 1:
Maize, which is utilized globally as 35 food, animal feed, and biofuel products, is one of the most important agro-economical crops. Despite the large production area for maize, extensive losses are common in China due to a range of biotic stressors, such as multiple insect injury. At present, although numerous genetic loci for multiple insect-resistant traits have been identified, little is known about genetic controls regarding DIMBOA content. In the present study, the best linear unbiased prediction (BLUP) values of DIMBOA content in two ecological environments across 310 maize inbred lines were calculated, and their phenotypic data 15 and BLUP values were used for marker-trait association analysis. The identified insect-resistant inbred lines, markers associated with DIMBOA content, and other multiple insect-resistant traits to ACB and CLA, as well as the corresponding candidate genes may be validated and utilized in MAS for the development of suitable varieties. The finding by itself is highly interesting. Overall, the present manuscript is well written, with the sound experimental designs and scientific data analysis. From all the above, I recommend that it can be accepted for publication in its present form.
Thanks for your positive comments and recognition of our work.
Thank you for your consideration.
English language and style are fine/minor spell check required.
Thanks for your positive comments. As suggested, the English language of the manuscript has been well modified by Charlesworth Author Services (https://www.cwauthors.com.cn/), and we provide the proof of English language polish. We then have re-submitted the manuscript.
Thank you for your consideration.
Reviewer 2:
The manuscript “Genetic Variation, DIMBOA Accumulation, and Candidate Genes Identification in Maize Multiple Insect-Resistance” is very interesting study.
Thanks for your positive comments and recognition of our work.
Thank you for your consideration.
Specific comments
- Kindly explain the abbreviation used in the abstract first time
Thanks for your positive comments. As suggested, we have explained the abbreviation used in the Abstract first time again, including “2,4-dihydroxy-7-methoxy-1,4-benzoxazin-3-one (DIMBOA)”, “best linear unbiased prediction (BLUP)”, “(auxin-upregulated RNAs (SAUR)”, and “v-myb avian myeloblastosis viral oncogene homolog (MYB)”. We then have re-submitted the manuscript.
Thank you for your consideration.
- It is better to show some quantitative data in the abstract.
Thanks for your positive comments. As suggested, we have revised the Abstract, i.e., “Maize seedlings contain high amounts of 2,4-dihydroxy-7-methoxy-1,4-benzoxazin-3-one (DIMBOA), and the effect of DIMBOA is directly associated with multiple insect-resistance against insect pests such as Asian corn borer and corn leaf aphids. Although numerous genetic loci for multiple insect-resistant traits have been identified, little is known about genetic controls regarding DIMBOA content. In this study, the best linear unbiased prediction (BLUP) values of DIMBOA content in two ecological environments across 310 maize inbred lines were calculated; and their phenotypic data and BLUP values were used for marker-trait association analysis. We identified nine SSRs that were signif-icantly associated with DIMBOA content, which explained 4.30–20.04% of the phenotypic variation. Combined with 47 original genetic loci from previous studies, we detected 19 hot loci, and approximately 11 hot loci (in Bin 1.04, Bin 2.00-2.01, Bin 2.03-2.04, Bin 4.00-4.03, Bin 5.03, Bin 5.05-5.07, Bin 8.01-8.03, Bin 8.04-8.05, Bin 8.06, Bin 9.01, and Bin 10.04 regions) supported pleiotropy for their association with two or more insect-resistant traits. Within the 19 hot loci, we identified 49 candidate genes including 12 controlling DIMBOA biosynthesis, six involving in sugar metabolism/homeostasis, two regulating peroxidases activity, 21 associated with growth and development [(auxin-upregulated RNAs (SAUR) family member and v-myb avian myeloblastosis viral oncogene homolog (MYB)], and seven involving several key enzyme activities (lipoxygenase, cysteine protease, restriction endonuclease, and ubiquitin-conjugating enzyme). The synergy and antagonism interactions among these genes formed the complex defense mechanisms induced by multiple insect pests. Moreover, sufficient genetic variation was reported for DIMBOA performance and SSR markers in the 310 tested maize inbred lines, and three highly (DIMBOA contents were 402.74-528.88 μg g-1 FW) and 15 moderate (DIMBOA contents were 312.92-426.56 μg g-1 FW) insect-resistant genotypes were major enriched in Reid group. These insect-resistant inbred lines can be used as parents in maize breeding programs to develop new varieties.” We then have re-submitted the manuscript.
Thank you for your consideration.
- Section 3 name should only be Discussion.
Thanks for your positive comments. As suggested, we have modified the corresponding content. We then have re-submitted the manuscript.
Thank you for your consideration.
- In discussion section, use the findings in this article to discuss, but not summarize the literature. There is need to enhance discussion section.
Thanks for your positive comments. As suggested, we have further condensed the corresponding content of Discussion section. We then have re-submitted the manuscript.
Thank you for your consideration.
- Conclusions are just repetition of results and needs to be shortened. Please provide some futuristic approaches and rewrite.
Thanks for your positive comments. As suggested, we have further reorganized and condensed the corresponding content of Conclusion section, namely “In summary, the DIMBOA confers significant resistance to ACB and CLA. In this study, SSR analysis revealed a wide genetic diversity in the 310 tested maize inbred lines from four type regions of Gansu Province of China, while Gansu Province is the largest maize seed production and breeding area in China. Population structure indicated that 294 inbred lines were successfully assigned to one or another group at a membership probability of ≥ 0.500. DIMBOA performance evaluation screened out three high and 15 moderate insect-resistant inbred lines, which can be used as parents in breeding programs to develop new maize varieties with multiple insect resistance. Using linkage mapping, we detected nine significant SSRs associated with DIMBOA content in both environments. We then combined the 47 original genetic loci for eight multiple insect-resistant traits from previous studies to detect 19 hot loci. Among them, 11 hot loci were located in Bin 1.04, Bin 2.00-2.01, Bin 2.03-2.04, Bin 4.00-4.03, Bin 5.03, Bin 5.05-5.07, Bin 8.01-8.03, Bin 8.04-8.05, Bin 8.06, Bin 9.01, and Bin 10.04 regions, and they were supported by pleiotropy for their association with two or more insect-resistant traits. Further, the 49 candidate genes involved in DIMBOA biosynthesis, sugar metabolism/homeostasis, and other multiple insect-resistance defense mechanisms in maize were identified in all 19 hot loci, and their highly interconnected network may form a complex maize multiple insect pests-induced defense mechanisms.” We then have re-submitted the manuscript.
Thank you for your consideration.
English language and style are fine/minor spell check required.
Thanks for your positive comments. As suggested, the English language of the manuscript has been well modified by Charlesworth Author Services (https://www.cwauthors.com.cn/), and we provide the proof of English language polish. We then have re-submitted the manuscript.
Thank you for your consideration.
Reviewer 3:
The manuscript entitled „Genetic variation, DIMBOA accumulation, and candidate genes identification in maize multiple insect-resistance“ represents a useful study aimed at an identification of loci associated with maize resistance to insect pests, namely Asian corn borer (Ostrinia furnacalis) and corn leaf aphid (Rhopalosiphum maidis). In the present study, 310 elite maize lines were used for phenotyping and marker-trait association analysis. The analysis of the 47 original genetic loci aimed at an identification of loci underlying differences in 2,4-dihydroxy-7-methoxy-1,4-benzoxazin-3-one (DIMBOA) content led to the identification of 19 hot loci encompassing 49 candidate genes including 12 controlling DIMBOA biosynthesis as well as other loci associated with other key metabolite pathways. Analysis of SSR markers associated with DIMBOA content led to the identification of 3 highly and 15 moderately resistant inbred lines which can be used as parents in maize breeding programs aimed at enhanced insect pests resistance. I have no major comment on the present manuscript. I have only a few major comments and a few formal comments on the present manuscript which are given below.
Thanks for your positive comments. As suggested, we have improved and modified the manuscript. We then have re-submitted the manuscript.
Thank you for your consideration.
Major comments:
Results: In Figure 3C providing a phylogenetic tree, an appropriate statistics, i.e., either a scale bar explaining the length of the branches or a number at branching point indicating the probability of the branching (bootstrap values per 1,000 replicates) have to be added.
Thanks for your positive comments. We used UPGMA cluster analysis via package of R (http://www.R-project.org/) to evaluate the multiple insect-resistance among 310 maize inbred lines in two ecological environments. As suggested, we have provided the corresponding appropriate statistics again, namely, “Figure 3. Evaluation of multiple insect-resistance among 310 maize inbred lines in both Zhangye (E1) and Longxi (E2) ecological environments by unweighted pair-group method with arithmetic means (UPGMA) cluster analysis, and their attributive group dissection. (A) Statistics for group number and frequency of each optimal group. (B) Attributive groups and their frequency distribution of three high insect-resistant and 15 moderate insect-resistant inbred lines; average of DIMBOA content for each attributive group (including Lüda red cob (LRC), Lancaster (Lan), P, and Reid group, respectively) in E1 and E2, respectively. (C) Evaluation of multiple insect-resistance among 310 inbred lines in two environments were performed using package of R (http://www.R-project.org/; accessed on 20 Apr. 2022) with UPGMA (The K value sets to 5), including type I (high insect-resistant lines), type II (moderate insect-resistant lines), type III (insect-resistant lines), type IV (moderate insect-susceptible lines), and type V (insect-susceptible lines), respectively.” We then have re-submitted the manuscript.
Thank you for your consideration.
Terminology: Line 293: Use only the heading 3. Discussion“ instead of „Discussion and Conclusionů since the part „5. Conclusions“ is given on lines 604-629.
Thanks for your positive comments. As suggested, we have modified the corresponding contents, namely “3. Discussion”. We then have re-submitted the manuscript.
Thank you for your consideration.
Materials and methods, lines 594-596: The symbols HB2 and HGE2 have to be explained not only by providing adequate formulae, but prviding adequate terminology (H means heredity??).
Thanks for your positive comments. As suggested, we have explained the ymbols HB2 and HGE2 in Materials and methods section, namely “The broad-sense heritability ( ) and genotype × environment interaction heritability ( ) values for DIMBOA contents under both environments were estimated as follows [54,66,67”. We then have re-submitted the manuscript.
Thank you for your consideration.
Formal comments:
Abstract, line 17: Terminology: Use the term „the phenotypic variation“ instead of „the phenotypic variance“.
Thanks for your positive comments. As suggested, we have corrected the word in Abstract section. We then have re-submitted the manuscript.
Thank you for your consideration.
Introduction, line 56: Add „a“ preceding the words „wide variation“, i.e., „a wide variation“.
Thanks for your positive comments. As suggested, we have modified the corresponding contents, namely “Fortunately, maize is a genetically diverse crop that exhibits a wide variation in its resistance to multiple insects [9,12,13]”. We then have re-submitted the manuscript.
Thank you for your consideration.
Results, line 139: Add a comma following the word „Thereby“ in the statement „Thereby, the genetic loci responsible for DIMBOA content in maize is necessary to detect.
Thanks for your positive comments. As suggested, we have modified the corresponding contents, namely “Thereby, the genetic loci responsible for DIMBOA content in maize is necessary to detect.”. We then have re-submitted the manuscript.
Thank you for your consideration.
Results: In Figure 3C providing a phylogenetic tree, an appropriate statistics, i.e., either a scale bar explaining the length of the branches or a number at branching point indicating the probability of the branching (bootstrap values per 1,000 replicates) have to be added.
Thanks for your positive comments. We used UPGMA cluster analysis via package of R (http://www.R-project.org/) to evaluate the multiple insect-resistance among 310 maize inbred lines in two ecological environments. As suggested, we have provided the corresponding appropriate statistics again, namely, “Figure 3. Evaluation of multiple insect-resistance among 310 maize inbred lines in both Zhangye (E1) and Longxi (E2) ecological environments by unweighted pair-group method with arithmetic means (UPGMA) cluster analysis, and their attributive group dissection. (A) Statistics for group number and frequency of each optimal group. (B) Attributive groups and their frequency distribution of three high insect-resistant and 15 moderate insect-resistant inbred lines; average of DIMBOA content for each attributive group (including Lüda red cob (LRC), Lancaster (Lan), P, and Reid group, respectively) in E1 and E2, respectively. (C) Evaluation of multiple insect-resistance among 310 inbred lines in two environments were performed using package of R (http://www.R-project.org/; accessed on 20 Apr. 2022) with UPGMA (The K value sets to 5), including type I (high insect-resistant lines), type II (moderate insect-resistant lines), type III (insect-resistant lines), type IV (moderate insect-susceptible lines), and type V (insect-susceptible lines), respectively.” We then have re-submitted the manuscript.
Thank you for your consideration.
Results, line 246: Modify the word form „Firstly“ to „First“ in the statement: „First, we collected 47 original genetic loci forACB/CLA-resistant traits…“
Thanks for your positive comments. As suggested, we have modified the corresponding contents, namely “First, we collected 47 original genetic loci for ACB-/CLA-resistant traits including DIMBOA content, tunnel length of corn borer (TL), aphid incidence rate (AIR), average aphid incidence grade (AIG), LFR, HO, tunnel length/number of holes of corn borer (TL/HO), and aphid resistance (AR) from previous studies (Table 2), and combined our results (Table 1) to construct a physical map (Figure 5).”. We then have re-submitted the manuscript.
Thank you for your consideration.
Line 253: Add a space betweenthe words „including“ and „12“.
Thanks for your positive comments. As suggested, we have modified the corresponding contents, namely “resulting in the identification of 49 candidate genes, including 12 controlling DIMBOA biosynthesis, six involving in sugar metabolism/homeostasis”. We then have re-submitted the manuscript.
Thank you for your consideration.
Results, Figure 6 legend, line 290: Modify the word form „significance“ to „significant“ in the statement „…indicate a significant difference“.
Thanks for your positive comments. As suggested, we have modified the corresponding contents, namely “Figure 6. The relative expression level of five candidate genes involved in DIMBOA biosynthesis, DIMBOA content observation, and their correlation in maize seedlings of RX20-1006 (high insect-resistant line) and T58 (insect-susceptible line) in Zhangye (E1) environment, Gansu Province, China. (A) The relative expression level (mean ± standard deviation) of the five candidate genes; different lowercase letters indicate a significant difference (p < 0.05) in gene expression. (B) DIMBOA content (mean ± standard deviation) in high insect-resisitant (RX20-1006) and insect-susceptible line (T58); different lowercase letters indicate a significance difference (p < 0.05) in DIMBOA content. (C) The correlational relationships between the relative expression level of five candidate genes and DIMBOA content; lines represent significant correlations (p < 0.05).” We then have re-submitted the manuscript.
Thank you for your consideration.
Discussion, line 299: Add „an“ preceding the word „increase“ in the statement „an increase of stalk lodging and bacterial/fungal infections…“
Thanks for your positive comments. As suggested, we have modified the corresponding contents, namely “It is well known that the plethora of ACB and CLA that either simultaneously or concurrently attack multiple maize parts, such as, newly hatched ACB feeds on whorl leaves and later instars tunnel into the stalk or the ear to feed on pith tissues or fresh kernels [25], resulting in a reduction of photosynthetic property, disruption of nutrient and water transport, an increase of stalk lodging and bacterial/fungal infections [25,26], and ultimately complicating harvesting practices and reducing grain yield and quality [27,28].” We then have re-submitted the manuscript.
Thank you for your consideration.
Discussion, line 355: Add either the word „lines“ or „materials“ in the statement: there is a great potential to obtain different multiple insect-resistant lines (materials) based on MAS application.
Thanks for your positive comments. As suggested, we have modified the corresponding contents, namely “Because of their clear differences in genetic effects and phenotypic variance in these loci, there is a great potential to obtain different multiple insect-resistant lines (materials) based on MAS application.” We then have re-submitted the manuscript.
Thank you for your consideration.
Line 373: Add a comma following „Thus“ in „Thus, these 11 hot loci have a pleiotropic effect…“
Thanks for your positive comments. As suggested, we have modified the corresponding contents, namely “Thus, these 11 hot loci have a pleiotropic effect on two to seven multiple insect-resistant traits, and Bin 1.04, Bin 2.00-2.01, Bin 4.00-4.03, Bin 5.05-5.07, Bin 8.04-8.05, Bin 8.06, and Bin 10.04 regions play important roles in conferring DIMBOA accumulation and other aspects of maize multiple-insect resistance to ACB and CLA.” We then have re-submitted the manuscript.
Thank you for your consideration.
Line 439: Add a comma between the words „and“ and „consistent with LOX funstion…“
Line 373: Add a comma following „Thus“ in „Thus, these 11 hot loci have a pleiotropic effect…“
Thanks for your positive comments. As suggested, we have modified the corresponding contents, namely “Lipoxygenase (LOX) plays critical roles in plant defense against multiple insect pests and pathogens [1,46,47]; and, consistent with LOX function”. We then have re-submitted the manuscript.
Thank you for your consideration.
Line 439: Write „LOX function“ without an apostroph in the scientific text.
Thanks for your positive comments. As suggested, we have modified the corresponding contents, namely “Lipoxygenase (LOX) plays critical roles in plant defense against multiple insect pests and pathogens [1,46,47]; and, consistent with LOX function”. We then have re-submitted the manuscript.
Thank you for your consideration.
Line 462: Add a comma between the words „Thereby“ and „sugar metabolism/homeostasis“.
Thanks for your positive comments. As suggested, we have modified the corresponding contents, namely “Thereby, sugar metabolism/homeostasis may play a positive role in maize defense response to ACB and CLA.” We then have re-submitted the manuscript.
Thank you for your consideration.
Discussion, line 495: Add „the“ in the statement: „In the current study,…“
Thanks for your positive comments. As suggested, we have modified the corresponding contents, namely “In the current study, a total of 748 alleles with a range of two to eight per locus were identified among 310 maize inbred lines using 186 polymorphic SSRs (Table S1).” We then have re-submitted the manuscript.
Thank you for your consideration.
Materials and methods, line 587: Replace the word „of“ with „in“ in the statement: „The average DIMBOA contents in 310 inbred lines from five biological replicates…“
Thanks for your positive comments. As suggested, we have modified the corresponding contents, namely “The average DIMBOA contents in 310 inbred lines from five biological replicates in each ecological environment were analyzed, respectively.” We then have re-submitted the manuscript.
Thank you for your consideration.
Materials and methods, lines 594-596: The symbols HB2 and HGE2 have to be explained not only by providing adequate formulae, but prviding adequate terminology (H means heredity??).
Thanks for your positive comments. As suggested, we have explained the ymbols HB2 and HGE2 in Materials and methods section, namely “The broad-sense heritability ( ) and genotype × environment interaction heritability ( ) values for DIMBOA contents under both environments were estimated as follows [54,66,67”. We then have re-submitted the manuscript.
Thank you for your consideration.
Conclusions, line 621: Add a comma between the words „thereby“ and „maize resistance“.
Thanks for your positive comments. As suggested, we have modified and improved the Conclusion section. We then have re-submitted the manuscript.
Thank you for your consideration.
Final recommendation: Accept after a minor revision.
Thanks for your positive comments. As suggested, we have modified and improved the Manuscript in turn. We then have re-submitted the manuscript.
Thank you for your consideration.
Sincerely,
Xiaoqiang Zhao professor
State Key Laboratory of Aridland Crop Science, Gansu Agricultural University
E-mail: zhaoxq3324@163.com

Reviewer 2 Report
The manuscript “Genetic Variation, DIMBOA Accumulation, and Candidate
Genes Identification in Maize Multiple Insect-Resistance” is very interesting study.
Specific comments
- Kindly explain the abbreviation used in the abstract first time
2. It is better to show some quantitative data in the abstract.
3. Section 3 name should only be Discussion.
4. In discussion section, use the findings in this article to discuss, but not summarize the literature. There is need to enhance discussion section.
5. Conclusions are just repetition of results and needs to be shortened. Please provide some futuristic approaches and rewrite.
Author Response

(The authors gave the same response as above.)

Reviewer 3 Report
Dear Authors,
Reviewer comments ijms-2121902
The manuscript entitled „Genetic variation, DIMBOA accumulation, and candidate genes identification in maize multiple insect-resistance“ represents a useful study aimed at an identification of loci associated with maize resistance to insect pests, namely Asian corn borer (Ostrinia furnacalis) and corn leaf aphid (Rhopalosiphum maidis). In the present study, 310 elite maize lines were used for phenotyping and marker-trait association analysis. The analysis of the 47 original genetic loci aimed at an identification of loci underlying differences in 2,4-dihydroxy-7-methoxy-1,4-benzoxazin-3-one (DIMBOA) content led to the identification of 19 hot loci encompassing 49 candidate genes including 12 controlling DIMBOA biosynthesis as well as other loci associated with other key metabolite pathways. Analysis of SSR markers associated with DIMBOA content led to the identification of 3 highly and 15 moderately resistant inbred lines which can be used as parents in maize breeding programs aimed at enhanced insect pests resistance.
I have no major comment on the present manuscript. I have only a few major comments and a few formal comments on the present manuscript which are given below.
Major comments:
Results: In Figure 3C providing a phylogenetic tree, an appropriate statistics, i.e., either a scale bar explaining the length of the branches or a number at branching point indicating the probability of the branching (bootstrap values per 1,000 replicates) have to be added.
Terminology: Line 293: Use only the heading „3. Discussion“ instead of „Discussion and Conclusionů since the part „5. Conclusions“ is given on lines 604-629.
Materials and methods, lines 594-596: The symbols HB2 and HGE2 have to be explained not only by providing adequate formulae, but prviding adequate terminology (H means heredity??).
Formal comments:
Abstract, line 17: Terminology: Use the term „the phenotypic variation“ instead of „the phenotypic variance“.
Introduction, line 56: Add „a“ preceding the words „wide variation“, i.e., „a wide variation“.
Results, line 139: Add a comma following the word „Thereby“ in the statement „Thereby, the genetic loci responsible for DIMBOA content in maize is necessary to detect.“
Results: In Figure 3C providing a phylogenetic tree, an appropriate statistics, i.e., either a scale bar explaining the length of the branches or a number at branching point indicating the probability of the branching (bootstrap values per 1,000 replicates) have to be added.
Results, line 246: Modify the word form „Firstly“ to „First“ in the statement: „First, we collected 47 original genetic loci forACB/CLA-resistant traits…“
Line 253: Add a space betweenthe words „including“ and „12“.
Results, Figure 6 legend, line 290: Modify the word form „significance“ to „significant“ in the statement „…indicate a significant difference“.
Discussion, line 299: Add „an“ preceding the word „increase“ in the statement „an increase of stalk lodging and bacterial/fungal infections…“
Discussion, line 355: Add either the word „lines“ or „materials“ in the statement: „there is a great potential to obtain different multiple insect-resistant lines (materials) based on MAS application.“
Line 373: Add a comma following „Thus“ in „Thus, these 11 hot loci have a pleiotropic effect…“
Lines 392-393: Use the word form „locus“ as a singular form instead of „loci“ which is a plura form, i.e., „locus 1“, „locus 3“, etc.
Line 439: Add a comma between the words „and“ and „consistent with LOX funstion…“
Line 439: Write „LOX function“ without an apostroph in the scientific text.
Line 462: Add a comma between the words „Thereby“ and „sugar metabolism/homeostasis“.
Discussion, line 495: Add „the“ in the statement: „In the current study,…“
Materials and methods, line 587: Replace the word „of“ with „in“ in the statement: „The average DIMBOA contents in 310 inbred lines from five biological replicates…“
Materials and methods, lines 594-596: The symbols HB2 and HGE2 have to be explained not only by providing adequate formulae, but prviding adequate terminology (H means heredity??).
Conclusions, line 621: Add a comma between the words „thereby“ and „maize resistance“.
Final recommendation: Accept after a minor revision.

Author Response

(The authors gave the same response as above.)
